# LET'S EXPLORE STEP BY STEP: GENERATING PROVABLE FORMAL STATEMENTS WITH DEDUCTIVE EXPLORATION

**Qi Liu, Kangjie Bao, Yue Yang, Xinhao Zheng, Renqiu Xia, Qinxiang Cao, Junchi Yan**[*]
School of Computer Science & School of Artificial Intelligence, Shanghai Jiao Tong University
Shanghai Innovation Institute
{purewhite,yanjunchi}@sjtu.edu.cn
https://github.com/Purewhite2019/dexploration_main

## ABSTRACT

Mathematical problem synthesis shows promise in resolving data exhaustion, contamination, and leakage for AI training and evaluation. Despite enormous efforts, an **expressiveness-validity-complexity trilemma** remains an open question. Existing methods either lack whole-process verifiability, are constrained to a particular domain, or are bounded by external models. This paper breaks the trilemma by proposing the framework of **DExploration** (*Deductive Exploration*), which formulates problem synthesis as a step-by-step exploration process instead of one-shot generation. Agents are equipped with three simple yet powerful atomic actions: *introducing* variables/hypotheses, *deducing* new facts, and *submitting* derived facts. The entire exploration process is formally verified by Lean 4, which encompasses most mathematical domains up to the research level. Once a conclusion is submitted, the framework outputs a formal statement with guaranteed provability, reducing the need for external models. To bootstrap training data for DExploration, we propose **Exploratory Transformation** to distill exploration trajectories from existing large-scale theorem-proving data. It rewrites formal proofs into a deductive style, parses dependencies among variables, hypotheses, and proof steps, then reassembles them into exploration trajectories by a topological order. Experiments validate the effectiveness and efficiency of our methods, achieving an improved success rate ($40.70\% \mapsto 54.52\%$), reduced token cost ($52.9\text{K} \mapsto 8.8\text{K}, 83\% \downarrow$), broader complexity and difficulty distributions, and Pareto optimality. In 2726 valid generations, three state-of-the-art provers fail on 60 (Pass@4) and 8 (Pass@64).

> *A mathematical problem should be difficult in order to entice us,*
> *yet not completely inaccessible, lest it mock at our efforts.*
>
> David Hilbert, *Mathematical Problems*

## 1 INTRODUCTION

State-of-the-art large language models (LLMs) demonstrate super-human reasoning capabilities (OpenAI et al., 2024a; DeepSeek-AI et al., 2025), especially in mathematical reasoning (Trinh et al., 2024; Huang & Yang, 2025; Chen et al., 2025). However, current evaluation pipelines suffer from data contamination (Golchin & Surdeanu, 2024; Choi et al., 2025), leakage (Xu et al., 2024), and static complexity (Zhu et al., 2023). LLM training is rapidly approaching the exhaustion of human-generated data (Villalobos et al., 2024; Yang et al., 2026). To continue pushing the boundaries of AI reasoning, there is a growing need for scalably generating fresh, valid, and complex problems as challenging evaluation benchmarks and high-quality training data (Team, 2025b).

Recent advances in mathematical problem synthesis show promise in addressing this need. However, existing methods suffer from a critical trilemma (Li et al., 2024b) across three core desiderata:

---

[*]Corresponding author. This work was in part supported by Scientific Research Innovation Capability Support Project for Young Faculty (U40) of the Ministry of Education of China, SRICSPYF-ZY2025019.

*expressiveness* (covering diverse mathematical domains), *validity* (generating provable and non-contradictory problems), and *complexity* (producing challenges that push AI limits). LLM-only methods (Luo et al., 2023; Tang et al., 2024; Zhao et al., 2025) synthesize questions and generate/filter answers using LLMs, which is error-prone. LLMs' capacity also limits the difficulty of the problems they generate. Domain-specific methods (Zhu et al., 2023; Trinh et al., 2024; Li et al., 2024b; Zhang et al., 2025) leverage deterministic algorithms to guarantee validity at the sacrifice of broad expressiveness. Formal-based methods combine LLMs and the Lean 4 environment (Moura & Ullrich, 2021), which spans up to research-level mathematics and can formally verify statement-proof pairs. These methods fall into two lines: Conjecture-prover methods (Poesia et al., 2024; Dong & Ma, 2025) directly generate formal statements and invoke LLM-based provers to attempt proofs; Autoformalizer-based methods (Huang et al., 2024; Leang et al., 2025) first synthesize informal problems and solutions, then autoformalize them into formal statements and proofs. Both paradigms inherit the diversity of LLMs and the expressiveness and verifiability of Lean. However, incorrect autoformalization undermines validity (Leang et al., 2025). The capabilities of autoformalizers and provers bound the complexity of synthesized problems. If the problems are too complex, formalizing or proving may easily fail, thus limiting the generation of truly challenging problems.

We propose **DExploration** (*Deductive Exploration*) to break through the complexity limit while maintaining expressiveness and verifiability. Instead of generating statements in one shot, agents explore the mathematical world step by step, with each move grounded in Lean. It supports three simple yet powerful atomic actions: *introducing* new variables or hypotheses, *deducing* new facts based on known context, and *submitting* any discovered fact as the conclusion. When a conclusion is submitted, we reconstruct a formal statement by concatenating the introduced hypotheses and the submitted fact. Then, a proof is extracted from the exploration trajectory, *ensuring the provability of the formal statement and reducing the need for external LLMs*.

As a novel framework, DExploration faces severe scarcity of training data. We propose **exploratory transformation** to transform statement-proof pairs of existing theorem-proving data into exploration trajectories. Given a statement-proof pair, we first transform the proof into a deductive manner (i.e., step-by-step logical inferences from hypotheses to conclusion). Next, we construct a *dependency graph* whose nodes are context variables and deductive steps, and whose edges represent the dependencies among them. We traverse the graph, following a topological order, and preferring deeper and deductive nodes. This traversal distills mathematical exploration intuitions into an exploration trajectory. Based on these data, we train the **DExplorer** agent as a proof-of-concept.

Experiments validate the effectiveness and efficiency of our methods. DExplorer demonstrates a significantly higher success rate ($40.70\% \mapsto 54.52\%$), magnitude of reduced token cost ($> 52.9$K $\mapsto$ 8.8K, $83\% \downarrow$), and better diversity (Rouge-L $0.186 \mapsto 0.173$). For formal reasoning, our generated problems exhibit greater complexity (Top-500 complexity $1231 \mapsto 1374$). For informal reasoning, our problems are more difficult (Top-500 difficulty $0.16 \mapsto 0.05$). Our methods also achieve Pareto optimality in terms of success rate. SOTA provers, including Goedel-Prover-V2-8B (Lin et al., 2025), DeepSeek-Prover-V2-7B (Ren et al., 2025), and Kimina-Prover-Distill-8B (Wang et al., 2025a), fail to prove 60 statements (Pass@4) and 8 (Pass@64) in total 2726 valid generated statements. This highlights the potential of our methods to push the boundaries of formal reasoners.

Ablation studies validate the necessity of our enhancements. Ablating DExploration, the resulting Conjecture-Prover system yields on-par performance compared to the strongest baseline. Ablating exploratory transformation results in a significantly lower success rate ($64.18\% \mapsto 52.10\%$) with on-par average token cost ($8841 \mapsto 8800$). The distribution of complexity also shifts downward ($515 \mapsto 470$ overall and $1374 \mapsto 1193$ for the Top-500).

In short, this paper aims to synthesize provable formal statements, benefiting both formal and informal reasoning. We identify the expressiveness-validity-complexity trilemma in existing problem synthesis methods and locate one of its root causes: dependency on external models for autoformalization or proof. The contributions of our work are summarized as follows.

1. **DExploration** (*Deductive Exploration*), a Lean 4-based framework for whole-process verifiable step-by-step exploration. The progressive exploration process enables unified generation of formal statements and proofs, ensuring the provability of generated statements and reducing the need for external provers. (Sec. 4.1)

2. **Exploratory transformation**, a pipeline to bootstrap initial training data for the **DExplorer** agent. A formal statement-proof pair is rewritten into deductive style, then parsed into a dependency graph, and finally reassembled into an exploration trajectory. (Sec. 4.2)

3. Extensive comparative experiments and ablation studies validate the effectiveness and efficiency of our methods. DExplorer demonstrates all-round improvements over baselines and ablations regarding formal reasoning, informal reasoning, efficiency, diversity, and Pareto-optimality. (Sec. 5)

## 2 RELATED WORKS

Trading off among the expressiveness-validity-complexity trilemma, existing literature on mathematical problem synthesis can be divided into three categories.

**LLM-based Methods.** Early pioneers in this category augment seed datasets (e.g., GSM8K) by prompting LLMs. WizardMath (Luo et al., 2023) uses Evol-Instruct to rewrite seed problems with different complexity or topic. MetaMath (Yu et al., 2023) augments problems by strategies including rephrasing, backward reasoning, and self-verification. AugGSM8K (Li et al., 2024a) applies various augmentation methods on GSM8K. Zhu et al. (2024) proposes to augment problems with a probing-judging multi-agent framework. Recent literature generates new problems and validates them with strong reasoners. MathScale (Tang et al., 2024) prompts LLMs to generate new problems conditioned on topics and knowledge points in seed datasets. ScaleQuest (Ding et al., 2025) fine-tunes an LLM to model the distribution of problems and samples from it. PromptCoT (Zhao et al., 2025) trains an LLM to generate both the rationales of problem construction and problems. While possessing the expressiveness of natural language, these methods face validity challenges due to error-prone LLM operations. The capability of quality evaluators bounds the quality of generated problems (Stroebl et al., 2024). Our methods differ by grounding the entire problem generation process in formal verification environments to ensure correctness.

**Domain-Specific Methods.** This literature introduces domain-specific symbolic methods to enhance validity. DyVal (Zhu et al., 2023) focuses on several template tasks and generates by sampling on a directed acyclic graph. Zhang et al. (2025) generates statements for two TCS problems. Trinh et al. (2024); Fu et al. (2025) focuses on geometry. Aygün et al. (2020); Puzis et al. (2006) focuses on first-order logic. Li et al. (2024b) pioneers to formalize and augment problems in the SMT-LIB language, then back-translate to natural language. Our methods are based on the Lean 4 (Moura & Ullrich, 2021) theorem-proving environment, whose expressiveness covers up to research-level mathematics (mathlib Community, 2020).

**Formal-based Methods.** These methods introduce formal theorem proving (FTP) for problem synthesis. One earlier line is *Autoformalizer-based methods*. MUSTARD (Huang et al., 2024) first generates a problem and a proof in natural language, then autoformalizes them into Lean 3 for verification. Leang et al. (2025) follows this paradigm and further proposes iterative autoformalization to mitigate formalization error. Another line is *Conjecture-Prover methods*, which directly generate formal statements and then attempt to generate their proofs. Poesia et al. (2024) samples conjectures from an LLM conditioned on target difficulty. STP (Dong & Ma, 2025) augments seed theorems conditioned on a related lemma. QDTSynth (Wang et al., 2025b) augments seed theorems by tactics. LeanConjecturer (Onda et al., 2025) synthesizes new statements based on seed files from Mathlib (mathlib Community, 2020). These works do not guarantee the validity of generated statements and require expensive LLM calls to generate formal proofs. Other works augment human-written Lean 4 proofs by symbolic equivalent/antecedent mutation (Wu et al., 2024) or forward reasoning (Rotella et al., 2025). Our methods differ in that we model problem generation as a series of fine-grained LLM-driven exploration steps in verified mathematics, thereby avoiding error-prone autoformalization (Leang et al., 2025) and reducing the need for proof generators.

## 3 PRELIMINARY

**Lean 4** (Moura & Ullrich, 2021) provides a theorem-proving environment that checks the correctness of statement-proof pairs. It is based on the Curry-Howard isomorphism (Howard et al., 1980), which views mathematical statements as types, similar to types in Python. A proof usually refers to a *tactic-style proof*, which consists of a sequence of *tactics* (like shell commands) that guide the Lean kernel to prove the statement. Examples of such statement-proof pairs can be found in Appendix H.

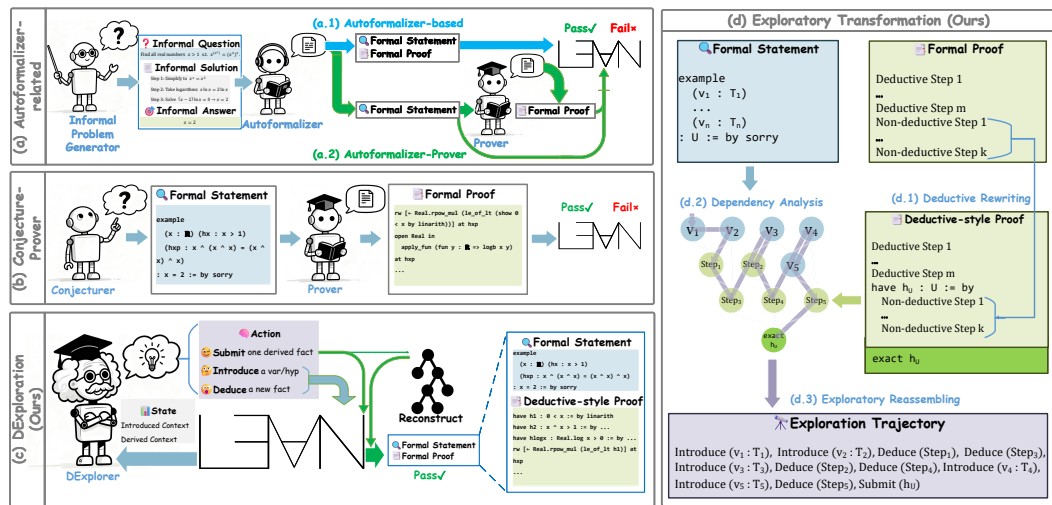

Figure 1: Pipelines of baselines (a.1, a.2, b), our DExploration (c), and exploratory transformation (d). Instead of directly synthesizing problems, **DExploration** supports agents in verifiably exploring the mathematical world step-by-step and submitting derived conclusions as problems. **Exploratory transformation** distills exploration trajectories from transforming theorem-proving data by (d.1) rewriting a formal proof into deductive-style, (d.2) constructing a dependency graph for conditions and proof steps, and (d.3) traversing the graph to reassemble an exploration trajectory.

**Whole-Proof Generation.** Given a formal statement, most state-of-the-art provers (Wang et al., 2025a; Ren et al., 2025; Lin et al., 2025) model proof generation as rejective sampling: repeatedly sample a whole proof candidate conditioned on the statement until the proof passes Lean typecheck.

**Tactic-Style Proving.** Notably, Lean supports writing proofs incrementally. It maintains a *proof state* initialized with the statement to prove. A proof state consists of a finite set of *proof goals* (things that we should prove) $\{\Gamma_i \vdash U_i\}_{i=1}^n$, where the *local context* $\Gamma_i$ is an ordered list of declarations (a list of given information), the target $U_i$ is a type (the conclusion to prove). A *declaration* can either be an assumption $(x : T)$ (declaring a variable or assuming a hypothesis) or a definition $(x := t : T)$. *Tactics* create, modify, or close proof goals. When all proof goals are closed, the proving succeeds.

## 4 METHOD

Approaches for mathematical problem posing can be divided into two directions (Silver, 1994; Liljedahl & Cai, 2021): *Backward* constructs or rewrites problems from particular knowledge points or seed problems. *Forward* is more genuinely discovering intriguing conclusions by exploring and investigating the mathematical world (Polya, 2021; Ernest et al., 2016; Elgrably & Leikin, 2021).

Inspired by forward posing, this section introduces **DExploration** (*Deductive Exploration*), a Lean-based framework where agents verifiably explore the math world step-by-step, and discover facts to synthesize problems. To bootstrap a **DExplorer** agent, **exploratory transformation** is proposed to distill exploration trajectories from existing formal statement-proof pairs.

### 4.1 FRAMEWORK: DEXPLORATION

DExploration formulates mathematical exploration as a deterministic Markov Decision Process.

**States.** We maintain an *exploration state* $S_t = (\Gamma_t, \Lambda_t) \in \mathcal{S}$, where the *introduced context* $\Gamma_t = [(v_i : T_i)]_{i=1}^n$ is an ordered list of all introduced variables and hypotheses; and the *deduced context* $\Lambda_t = [(h_i : H_i)]_{i=1}^m$ is an ordered list of all derived intermediate facts. $S_0$ is initialized as $([\,],[\,])$.

**Actions.** To enable active exploration, three basic but powerful actions should be supported:

- Introduce$(v : T)(\Gamma, \Lambda) = (\Gamma \circ [(v : T)], \Lambda)$. Add a new variable or hypothesis $(v : T)$ to $\Gamma$.

- Deduce$(h : H)(\Gamma, \Lambda) = (\Gamma, \Lambda \circ [(h : H)])$. Deduce a fact $H$ based on introduced variables, hypotheses, and previously derived facts. It should propose the assertion $H$ and prove $\Gamma \circ \Lambda \vdash H$.
- Submit$(h : H)(\Gamma, \Lambda) = \Gamma \vdash H$. Submit the fact $H$ as the target conclusion, reconstruct a statement $\Gamma \vdash H$ and its proof, and end the episode.

**Implementation.** We concretize DExploration in Lean 4 (Moura & Ullrich, 2021). The union of the introduced context $\Gamma$ and the deduced context $\Lambda$ corresponds to Lean's proof state $\{\Gamma \circ \Lambda \vdash \texttt{False}\}$. To differentiate from $\Lambda$, the introduced context $\Gamma$ is also maintained externally.

Actions are implemented using tactics. Introduce$(v : T)$ is concretized with `obtain` $\langle v, \_ \rangle :$ $\exists (v : T), \text{True} := \texttt{sorry}$, which introduces a free variable or hypothesis $(v : T)$ without actual proof. The placeholder $(\_ : \text{True})$ is then removed to avoid context rot. The name of the introduced variable $v$ should not contradict with existing variables. Care should be taken to avoid introducing a hypothesis $(v : T)$ that leads to a contradiction. If $\Gamma \circ [v : T] \vdash \texttt{False}$ holds, anything $\Gamma \circ [v : T] \vdash U$ follows. We add an explosion check after introducing a hypothesis: execute a light-weighted heuristic to prove `False`. If it succeeds, the introducing step is rejected.

Strict implementation of Deduce$(h : H)$ only allows the `have`-tactic, which proposes and proves one hypothesis $H$. For stronger expressiveness and affinity to traditional theorem-proving, we relax the restriction to *deductive tactics*, which covers tactics that 1) does not create extra goals, only modifies the current goal; 2) does not modify the target of the goal; 3) does not modify variables (in the `Type` universe) in local context. For example, `have` and `apply at` are deductive tactics, while `constructor` and `apply` are not. More discussions can be found in Appendix A.

Submit$(h : H)$ is implemented externally. The output statement $\Gamma \vdash H$ is guaranteed to be provable. A corresponding proof can be reconstructed by concatenating all deducing steps and transforming the submission step into an `exact` tactic. Proofs and discussions can be found in Appendix A.

Therefore, DExploration enables whole-process verified math exploration and unified generation of statements and proofs, reducing the need for extra provers. However, LLM-based agents for DExploration face extreme scarcity of training data.

## 4.2 DATA BOOTSTRAPPING: EXPLORATORY TRANSFORMATION

To construct training data for DExploration, we distill human creativity and intuition from existing large-scale theorem-proving data. We propose *exploratory transformation* to transform off-the-shelf statement-proof pairs into exploration trajectories, which consists of three main steps: deductive rewriting, dependency analysis, and exploratory reassembling.

**Deductive Rewriting**. Given a statement-proof pair $(\Gamma \vdash U, [s_i]_{i=1}^m)$, where $\Gamma = [(v_i : V_i)]_{i=1}^n$, we first transform the proof $[s_i]_{i=1}^m$ into a deductive-style proof. Suppose the first non-deductive tactic is $s_k (0 \leq k \leq m)$, we obtain a series of deducing tactics $[s_i]_{i \leq k}$. The remaining tactics are wrapped into two deductive tactic: $s_{-2} := \texttt{have } h_U : U := \texttt{by } [s_i]_{k < i \leq m}$, and $s_{-1} = \texttt{exact } h_U$. Then, deductive tactics $[s_i]_{i \leq k} \circ [s_{-2}, s_{-1}]$ forms a deductive proof.

**Dependency Analysis.** We further analyze the relations among variables in $\Gamma$ and deductive steps $[s_i]_{i=1}^m$. A variable $(v : T)$ in $\Gamma$ *depends* on another $(v' : T')$ if $v'$ occurs in $T$. Deductive steps manipulate the context of the proof goal. Therefore, if a step $s_i$ uses variables $V_-(s_i) := \{(v_k : T_k)\}_{k=1}^n$ in its execution and creates $V_+(s_i) := \{(v_k : T_k)\}_{k=1}^m$, it *depends* on all variables in $\Gamma \cap V_-$. Moreover, all subsequent steps $s_j$ that $V_-(s_j) \cap V_+(s_i) \neq \emptyset$ *depends* on $s_i$, because removing $s_i$ from $[s_k]_{k \leq j}$ makes $s_j$ fail to execute. As visualized in Fig. 1(d.2), the dependency relations form a directed acyclic graph (DAG) where each node represents a variable in $\Gamma$ or a deductive step, each edge represents a dependency.

**Exploratory Reassembling.** During exploration, a mathematician prefers to derive as many conclusions as possible, and only introduce new hypotheses when necessary (Elgrably & Leikin, 2021). We distill this depth-preference by reassembling nodes in the dependency graph into an exploration trajectory. Let $\text{dep}(s)$ denote the set of nodes that a node $s$ depends on. The *reasoning depth* of $s$ is

$$d(s) = \begin{cases} \max_{s' \in \text{dep}(s)} d(s') + 1 & \text{dep}(s) \neq \emptyset \\ 0 & \text{dep}(s) = \emptyset \end{cases} \tag{1}$$

Formally, $d(s)$ represents the level of topological generation for $s$. Intuitively, $d(s)$ reflects the depth of exploration: $s$ can be reached from basic hypotheses with $d(s)$-times reasoning. We conduct a topological sort on the dependency graph while prioritizing available nodes that 1) have the highest reasoning depth, and 2) are not introducing steps. To obtain an exploration trajectory from the traversal path, we substitute all nodes of context variables $(v : T)$ with introducing steps Introduce$(v : T)$, and replace all nodes of deductive tactic $s$ with deducing steps $s$.

## 5 EXPERIMENT

This section validates the effectiveness and efficiency of the proposed methods. Sec. 5.1 introduces comparative and ablative methods for evaluation. Sec. 5.2 details the evaluation settings. Sec. 5.3 reports and analyzes the experiment results. All experiments share identical training recipe and sampling parameters, as detailed in Appendix E. Prompt templates are in Appendix G. Lean 4 environment details are in Appendix F. Details for benchmarking informal reasoning are in Appendix B.

### 5.1 EVALUATED METHODS

Two main paradigms, Autoformalizer-based and Conjecturer-Prover, are evaluated. We form a stronger baseline, Autoformalizer-Prover, by chaining SOTA informal problem synthesis methods, autoformalizers, and provers. For ablation, Conjecturer-Prover shares the training dataset with DExplorer while ablating DExploration. DExplorer (Staged) ablates the exploratory transformation by forcing the exploration to follow the order of introducing, deducing, and submitting.

**Autoformalizer-based** methods (Huang et al., 2024; Leang et al., 2025) (Fig. 1(a.1)) prompt generalist LLMs to generate informal problems and proofs conditioned on randomly sampled math concepts. Then, they prompt LLMs to autoformalize the informal data into formal code. We adopt published data from **MUSTARD** (Huang et al., 2024), which consists of 28,316 Lean 3 files generated by GPT-4 (OpenAI et al., 2024b), where 5,866 are typechecked. We utilize Mathport (leanprover-community, 2024) to migrate them to Lean 4, yielding 3,794 formal statement-proof pairs.

**Autoformalizer-Prover** methods (Fig. 1(a.2)) first synthesize an informal question-answer pair, call an autoformalizer to formalize it into a formal statement, and then call LLM provers to generate proofs. We adopt published data from **ScaleQuest-Math** (Ding et al., 2025), **PromptCoT-DS**, and **PromptCoT-QwQ** (Zhao et al., 2025) as the informal data source. We adopt Kimina-Autoformalizer-7B (Wang et al., 2025a) as the autoformalizer. Three SOTA LLM provers: Goedel-Prover-V2-8B (Lin et al., 2025), DeepSeek-Prover-V2-7B (CoT mode) (Ren et al., 2025), and Kimina-Prover-Distill-8B (Wang et al., 2025a) are invoked to generate at most $N_p = 4$ proof candidates. If at least one passes Lean typecheck, the statement is proven.

**Conjecturer-Prover** methods (Poesia et al., 2024; Dong & Ma, 2025) (Fig. 1(b)) use an LLM conjecturer to generate formal statements, and call provers to prove. For a fair comparison, we fine-tuned the conjecture from Goedel-Prover-V2-8B (Lin et al., 2025) using the proven formal statements in NuminaMath-Lean (Wang et al., 2025a). Generated conjectures are proven using the identical prover settings to the Autoformalizer-Prover experiment. Compared with DExplorer, Conjecturer-Prover directly generates the entire formal statement while sharing the same training set and base model. Therefore, it also works as an ablation of DExploration.

**DExplorer.** (Fig. 1(c)) We apply exploratory transformation on the NuminaMath-Lean (Wang et al., 2025a) dataset, obtaining 39,509 exploration trajectories to fine-tune Goedel-Prover-V2-8B (Lin et al., 2025) as the agent. In each episode, the agent generates at most $N_s = 80$ steps.

**DExplorer (Staged).** Based on DExplorer, we restricted the exploration to be staged: the agent first generates a series of introducing steps. Once it generates the first deducing step, it is then constrained to only generate deducing or submitting steps. We keep the step limit $N_s = 80$. This experiment ablates the exploratory transformation by disabling interleaved deducing and introducing.

### 5.2 EVALUATION SETTING

For all methods excluding MUSTARD, we run $5,000$ episodes to generate formal statements and proofs. For MUSTARD, all 3794 formal statement-proof pairs participate in the evaluation. The

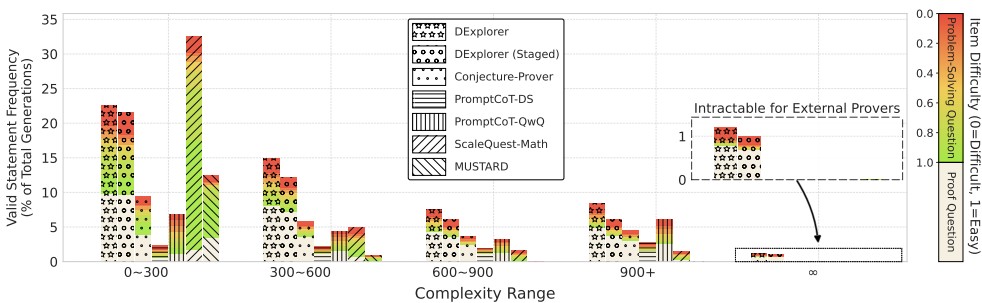

Figure 2: Frequency of valid statement across complexity ranges, normalized to total generations. The portion of informalized problem-solving questions and proof questions is colored with item difficulty (lower is more difficult). DExplorer outperforms all baselines in generating difficult statements ($\geq 300$ complexity or $\leq 0.75$ item difficulty).

number of output tokens during generation is recorded. For fair comparisons of proof lengths, valid statements in MUSTARD and our methods are re-proven using identical prover settings to those in the Autoformalizer-Prover experiment.

Moreover, we check the contradiction and validity of each statement. A statement $\Gamma \vdash U$ is *contradictory* if $\Gamma \circ [h_U : U] \vdash \texttt{False}$ holds[1]. Intuitively, this occurs when conditions are mutually contradictory. A *valid* statement is proven and not contradictory. We first attempt to prove the **satisfiability** $[\,] \vdash \exists \Gamma, \texttt{True}$ with the three provers, each generating one proof candidate. If proven, the original statement is not contradictory. Otherwise, we try directly proving the **contradiction** $\Gamma \circ [h_U : U] \vdash \texttt{False}$ with the three provers, each tries 4 times. If proven, the original statement is contradictory. Otherwise, it is viewed (but not guaranteed) as not contradictory. For the light-weighted explosion check in DExploration, we use 1) Aesop (Limperg & From, 2023), a heuristic automation, and 2) DeepSeek-Prover-V2-7B (Non-CoT mode) (Ren et al., 2025) to sample 1 proof.

We also explore possibilities to facilitate informal math reasoning. With 5-shot prompting DeepSeek-V3.1 (DeepSeek-AI, 2024), all valid formal statements are categorized and informalized into problem-solving questions or proof questions in natural language. We uniformly sample 100 informalized problem-solving questions and invited human experts for a quality check. 97 of them are correct, while only 3 are incorrect due to misformulating existence proof problems as problem-solving questions. We then evaluate the difficulty of the synthesized problem-solving questions with a broad spectrum of open-source models, including Qwen3-8B (Team, 2025b), Phi-4-mini-instruct (Microsoft et al., 2025), OLMo-2-1124-7B-Instruct (OLMo et al., 2024), Mistral-7B-Instruct-v0.3 (Jiang et al., 2023), Llama-3.2-3B-Instruct (Grattafiori et al., 2024), and Llama-3.1-8B-Instruct (Grattafiori et al., 2024).

## 5.3 RESULTS AND DISCUSSIONS

### 5.3.1 MAIN RESULTS

The overall experiment results are summarized in Tab. 1. Case studies of generated statements of DExploration can be found in Appendix H. We mainly evaluate the methods regarding the following aspects. Rubrics-based evaluation of mathematical value is in Appendix D, where DExploration shows superior top-500 scores and on-par overall averages with best baselines.

**Formal Reasoning.** The complexity of formal statements is measured with proof lengths (Huang et al., 2024) (w/o comments and whitespaces). DExplorer demonstrates a clear advantage over baselines on the average complexity of top-500 statements (Cplx. 500). However, PromptCoT-QwQ and PromptCoT-DS show higher overall average complexity. We break down the generation frequency over complexity ranges in Fig. 2, which demonstrates that our method outperforms all baselines and ablations in generating problems with $> 300$ complexity. ScaleQuest generates the most problems in the $[0, 300)$ complexity range, followed by our methods. Notably, 60 statements

---

[1] Otherwise, either $\Gamma \vdash U$ does not hold (unprovable) or $\Gamma \vdash \texttt{False}$ holds. In the latter case, the conclusion $U$ is doubtful since $\texttt{False} \vdash A$ holds for any assertion $A$ (explosion principle)

Table 1: Comparative and ablative experiment results. {**All**,**Proven**,**Contra.**,**Valid.**, **Intract.**}: numbers of {all / proven / contradictory / valid / valid and unprovable by external provers} generated statements; **Token Cost**: average output token numbers for each valid statements. {- / >} indicates {missing / underestimated}; {**Cplx.500**,**Cplx.**}: {Top-500 / Overall} average complexity (i.e., proof length) of valid statements; {**Diff.500**,**Diff.**}: {Top-500 / Overall} average item difficulty of informalized problem-solving questions (lower is more difficult); **Rouge-L**: the Rouge-L scores of informalized questions and answers (lower is more diverse). Bold numbers emphasize best values; Underlined numbers emphasize second-best values.

| Method | All | Proven↑ | Contra.↓ | Valid↑ | Intract.↑ | Token Cost↓ | Cplx.500↑ | Cplx.↑ | Diff.500↓ | Diff↓ | Rouge-L↓ |
|---|---|---|---|---|---|---|---|---|---|---|---|
| **MUSTARD** | 28316 | 3794 | **3** | 3791 | - | - | 335 | 119 | 0.16 | 0.80 | 0.202 |
| **PromptCoT-QwQ** | 5000 | 1189 | 729 | 1024 | - | > 172,927 | 1231 | **721** | 0.30 | 0.45 | 0.187 |
| **PromptCoT-DS** | 5000 | 678 | 927 | 457 | - | > 441,738 | 786 | **786** | 0.40 | **0.40** | 0.186 |
| **ScaleQuest-Math** | 5000 | 2369 | 1591 | 2035 | - | > 52,915 | 599 | 217 | 0.28 | 0.76 | 0.220 |
| **Conjecture-Prover** | 5000 | 1589 | 1817 | 1164 | - | 187,128 | 1072 | 578 | 0.51 | 0.52 | 0.174 |
| **DExplorer (Staged)** | 5000 | 2605$_{(+1016)}$ | 269$_{(-1548)}$ | 2340$_{(+1176)}$ | 50 | **8,800**$_{(0.05\times)}$ | 1193$_{(+120)}$ | 470$_{(-108)}$ | 0.09$_{(-0.42)}$ | 0.47$_{(-0.05)}$ | **0.173**$_{(-0.001)}$ |
| **DExplorer** | 5000 | **3209**$_{(+1620)}$ | 497$_{(-1320)}$ | **2726**$_{(+1562)}$ | **60** | 8,841$_{(0.05\times)}$ | **1374**$_{(+301)}$ | 515$_{(-63)}$ | **0.05**$_{(-0.46)}$ | 0.47$_{(-0.05)}$ | **0.173**$_{(-0.001)}$ |

(2.20% valid statements) generated by our methods have $\infty$-complexity, i.e., not proven by any of the SOTA provers. We further evaluate all the provers with Pass@64. 8 of the statements (0.29% valid statements) remain intractable. This demonstrates our method's potential to surpass existing provers and uncover more challenging conjectures while ensuring provability.

**Informal Reasoning.** We use item difficulty (Crocker & Algina, 1986), i.e., the proportion of LLMs who answer a question correctly (Pass@1), to measure the *difficulty* of each informalized problem-solving question. Due to space limitations, we report the average difficulty of the top-500 most difficult questions and overall questions in Tab. 1. Appendix B details the performance of all evaluated models. Our top 500 problems are harder than AIME24, and the overall problems are more difficult than MATH500. Difficulty shares similar trends with complexity. The top-500 difficult problems of DExplorer are significantly harder than baselines (0.05 vs. 0.16). However, the overall problems are slightly easier than PromptCoT-QwQ and PromptCoT-DS (0.45 and 0.40 vs. 0.47). As visualized in Fig. 2, among all complexity groups, our method generates more difficult problems (colored red) than any baselines and ablations. The seemingly easier overall difficulty is due to our method's capability to generate simple problems.

**Diversity.** Following Huang et al. (2024), we compute ROUGE-L (Lin, 2004) on the informalized questions, answers, and solutions to measure the diversity of generation. For each experiment, we compute the average pair-wise ROUGE-L scores over 400 uniformly sampled data points. We report the average results from 5 independent runs, resulting in a standard deviation of $\leq 0.003$ for each method. All methods' Rouge-L scores are below 0.25, indicating high generation diversity. Our methods consistently demonstrate lower Rouge-L (higher diversity) than baselines (0.173 vs. 0.186 for PromptCoT-DS). Although fine-tuned from the same model on the same datasets, DExplorer is slightly more diverse than Conjecturer-Prover (0.173 vs. 0.174). This improvement may be explained by the fact that Conjecturer-Prover generates fewer high-complex and difficult statements, as demonstrated in Fig. 2.

**Efficiency.** We measure generation efficiency by *proven rate*, *contradictory rate*, and *success rate* (the portion of proven, contradictory, and valid statements among all generated ones), and *token cost*(the average number of generated tokens for each valid statement). DExplorer still demonstrates a drastically higher proven rate (64.18% vs. 47.38% of ScaleQuest-Math) and success rate (54.52% vs. 40.70% of ScaleQuest-Math), a lower contradictory rate (9.94% vs. 14.58% of PromptCoT-QwQ) with magnitude-lower token cost (8.8K vs. 187K of Conjecturer-Prover). Notably, the reported cost of MUSTARD does not include the cost of generating informal problems, informal proofs, and autoformalization. The reported costs of PromptCoT-QwQ, PromptCoT-DS, and ScaleQuest-Math also do not include the cost of generating informal question-answer pairs. Therefore, the actual cost of the baselines is significantly higher than the reported cost. As shown in Fig. 2, the generated statements of our method span a wide spectrum of complexity and difficulty, whereas ScaleQuest-Math and MUSTARD concentrate on generating easier problems, PromptCoT-DS and PromptCoT-QwQ focus on harder problems. DExplorer generates more difficult problems than baselines and ablations.

**Pareto-Optimality.** By adjusting the step limit $N_s$ among $\{4, 8, 16, 32, 80\}$ of DExplorer and the proof limit $N_p$ among $\{1, 2, 4, 8, 12\}$ for prover-based methods, we further investigate the perfor-

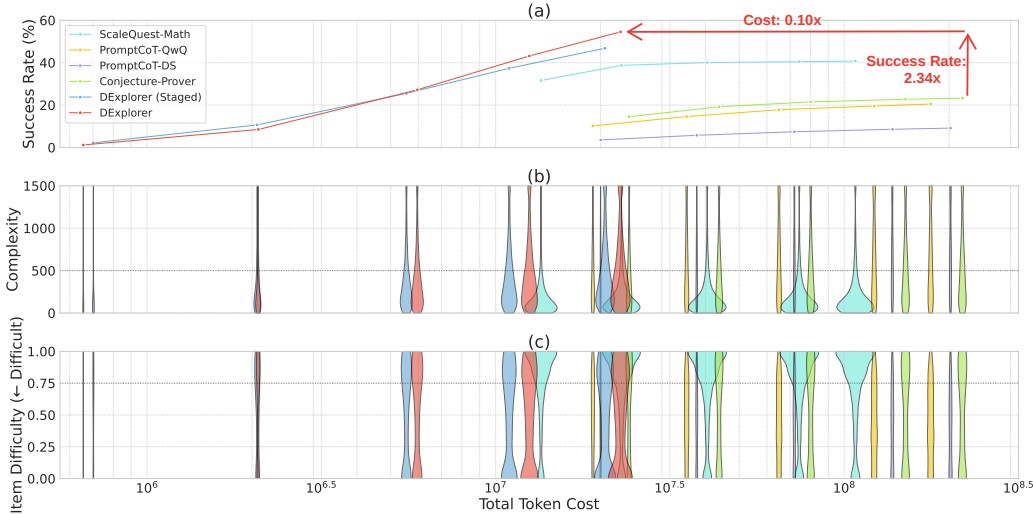

Figure 3: Results of cost-controlled experiments. **(a)** Success rate across total token costs. **(b)** Distribution of formal statements' complexity across total token costs. Widths are proportionally scaled to the success rate. **(c)** Distribution of informal problem-solving questions' item difficulty across total token costs. Lower indicates more difficult. Widths are proportionally scaled to the number of valid statements.

mance of different methods under different computational budgets. The success rate curves across total token costs are visualized in Fig. 3(a), where DExplorer demonstrates pareto-optimality compared with all baselines. Even under an unfair comparison, where the token costs of baseline methods are significantly underestimated, DExplorer requires orders of magnitude lower computational cost to reach the same level of success rate: $0.13\times$ that of ScaleQuest-Math, and $< 0.1\times$ that of PromptCoT and Conjecture-Prover. The distribution of complexity to formal methods and difficulty to informal methods across different total token costs is shown in Fig. 3(b) and Fig. 3(c), respectively. DExplorer shows strong cost-effectiveness over all baselines by generating more samples at all complexity levels, except ScaleQuest-Math. The distribution of ScaleQuest-Math concentrates on simpler statements with complexity $< 500$ or item difficulty $> 0.75$. Ours is more balanced and generates more than any baseline for complexity $> 500$ or item difficulty $< 0.25$ (harder), with a $0.09\times$ cost of Conjecturer-Prover. With increasing total cost, our methods exhibit clear distribution shifts towards more complex and difficult problems, which are discussed in Appendix C and illustrated in Fig. 4, 5.

**Ablation.** Comparisons to Conjecturer-Prover shows all-around superiority of DExplorer, including higher proving rate ($31.78\% \mapsto 64.18\%$), lower contradictory rate ($36.34\% \mapsto 9.94\%$), higher success rate ($23.28\% \mapsto 54.52\%$), lower token cost ($187K \mapsto 8.8K$), more complex generations (Fig. 2), lower average item difficulty ($0.52 \mapsto 0.47$, lower is more difficult), and lower Rouge-L ($0.174 \mapsto 0.173$, lower is more diverse), and better cost-effectiveness (Fig. 3(a)-3(c)). Note that Conjecturer-Prover and DExplorer are fine-tuned using the same model, dataset, and hyperparameters. This ablation suggests that the advantages of DExplorer are not attributed to the dataset but to the reasoning architecture. By decomposing mathematical conjecturing into step-by-step exploration, the logical gaps between conditions and conclusions are narrowed, and the requirement of external provers is eliminated.

Ablative experiments of the DExplorer (Staged), which restricts the DExplorer to staged exploration, also show inferior or on-par performance compared to the original method across all aspects. However, as in Fig. 3(a), in an extremely low budget scenario where the total cost $< 10^{6.5}$ (step limit $N_s < 8$), the ablation slightly outperforms the main method on success rate. This may be explained by the fact that staged exploration prefers shallower exploration since it prohibits introducing new variables and hypotheses upon deduction.

## 6 CONCLUSION

This paper aims to synthesize provable formal statements with efficiency, complexity, difficulty, and diversity. We identify an expressiveness-validity-complexity trilemma in existing problem synthesis methods and locate one root cause: reliance on external reasoners. To break the trilemma, we propose an end-to-end verified framework, **DExploration** (*Deductive Exploration*). Instead of directly synthesizing problems, DExploration enables agents to explore the mathematical world and discover intriguing conclusions by supporting three atomic actions: *introducing*, *deducing*, and *submitting*. Once a conclusion is submitted, a provable statement is constructed by composing the introduced conditions and the submitted conclusion. We design **Exploratory Transformation** to address data scarcity by transforming existing statement-proof pairs into exploration processes. Comparisons to strong baselines and thorough ablations validate the effectiveness and efficiency of our methods.

### REPRODUCIBILITY STATEMENT

Our research aims to extend the applications of formal theorem proving to broader domains by proposing a framework for effectively and efficiently synthesizing provable formal statements, as well as a data bootstrapping method based on abundant theorem-proving data. We appreciate and affirm the value of reproducibility in scientific research and therefore summarize the details of methods, data, models, and experiments as follows:

- The Lean 4 dependencies and environment settings are detailed in Appendix F;
- The training recipe, inference hyperparameters, and generating settings are detailed in Appendix E;
- Details about benchmarking informal reasoning, including the evaluated models, the evaluation toolkit, the pipeline for quality check, human evaluation result, and detailed evaluation results are in Appendix B;
- The Prompt templates for DExplorer, conjecturer, informalization, and informal reasoning are in Appendix G;
- Evaluated methods and evaluating setting is detailed in Sec. 5.2.

Moreover, we will upload all generated statements and evaluation results as supplementary materials. Code, data, and models will be available after acceptance.

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

## A  FORMAL PROOF OF RECONSTRUCTION CORRECTNESS

Let $[s_1, s_2, \ldots, s_T]$ be the sequence of exploration steps, where $s_T = \text{Submit}(h_U : U)$. Let $\Gamma_i \vdash$ `False` denote Lean proof state before step $s_i$, with $\Gamma_1 = \emptyset$. Vacuous steps $s_k$ that $\Gamma_{k+1} \setminus \Gamma_k = \emptyset$ are neglected. The indices $\{1, 2, \ldots, T-1\}$ are partitioned into two disjoint lists based on the type of steps: $I$ (Introduction) and $D = [D_1, D_2, \ldots, D_m]$ (Deduction). For any $k \in \{1, 2, \ldots, m\}$, the step $s_{D_k}$ is a deductive tactic that transforms proof state $\Gamma_{D_k} \vdash$ `False` to $\Gamma_{D_k+1} \vdash$ `False`. By the definition of Submit(), we have $(h_U : U) \in \Gamma_{D_m+1}$. For $i \in I$, $s_i$ introduces a new variable or hypothesis $\Gamma_{i+1} = \Gamma_i \cup \{(v_i : T_i)\}$.

**Statement Reconstruction.** The reconstructed statement is formalised as $\Gamma^\circ \vdash U$, where $\Gamma^\circ = \{(v_k : T_k)\}_{k \in I}$ represents the collection of all introduced variables and hypotheses throughout the exploration.

**Proof Reconstruction.** The reconstructed proof script is the sequence:

$$[\text{Encapsulate}(s_{D_k}, \Gamma^\circ, \Gamma_{D_k}, \Gamma_{D_k+1})]_{k=1}^m \circ [\texttt{exact } h_U]$$

where the Encapsulate operation wraps the deductive tactic to ensure context consistency.

**The Encapsulate Operator.** Given a deductive step $s$ that transforms proof state $\Gamma \vdash$ `False` to $\Gamma' \vdash$ `False`, the operator $\text{Encapsulate}(s, \Gamma^\circ, \Gamma, \Gamma')$ represents the following Lean tactic sequence. It temporarily restricts the proof state to match $\Gamma$, executes $s$, and then restores the wider context.

Denote the sets of declarations involved as:

- *Future Variables*: $\Gamma^\circ \setminus \Gamma = \{(v_i : T_i)\}_{i=1}^n$ (variables introduced after the step $s$).
- *New Deductions*: $\Gamma' \setminus \Gamma = \{(h_i' : H_i')\}_{i=1}^p$.
- *Consumed Hypotheses*: $\Gamma \setminus \Gamma' = \{(h_i : H_i)\}_{i=1}^q$ (hypotheses consumed/cleared by $s$).

```
have _H_DEDUCTION_ENCAPSULATION := by
  -- The proof target U is hidden
  revert v₁ v₂ ... vₙ
  -- Future Variables Γ°∖Γ are hidden
  have _H_DEDUCTION_ENCAPSULATION := by
    -- The context is now identical to Γ
    s
```

```
    -- The context is now identical to Γ′
    exact (show (H′₁ ∧ H′₂ ∧ ... ∧ H′ₚ) by exact ⟨h′₁, h′₂, ..., h′ₚ⟩)
  intros
  exact _H_DEDUCTION_ENCAPSULATION
-- The derived facts are consolidated to '_H_DEDUCTION_ENCAPSULATION'
clear h′₁ h′₂ ... h′ₚ h₁ h₂ ... h_q
rcases _H_DEDUCTION_ENCAPSULATION with h′₁, h′₂, ..., h′ₚ
-- Apply changes from Γ to Γ′ in the current proof state
```

If $p = 1$, the `rcases` simplifies to a `rename′`.

```
rename′ _H_DEDUCTION_ENCAPSULATION => h′₁
```

Moreover, if $\Gamma^\circ \setminus \Gamma = \emptyset$, the double `have` `_H_DEDUCTION_ENCAPSULATION` encapsulation is not required.

```
have _H_DEDUCTION_ENCAPSULATION := by
  -- The proof target U is hidden
  -- The context is now identical to Γ
    s
  -- The context is now identical to Γ′
  exact (show (H′₁ ∧ H′₂ ∧ ... ∧ H′ₚ) by exact ⟨h′₁, h′₂, ..., h′ₚ⟩)
-- The application of changes from Γ to Γ′ is identical to above.
...
```

**Correctness Proof.** We prove by induction that for any $k \in \{1, \ldots, m\}$, the proof state after executing the first $k$ encapsulated steps from the initial proof state $\Gamma^\circ \vdash U$ is $\Gamma^{(k+1)} \vdash U$, where

$$\Gamma^{(k+1)} = \{(v_i : T_i)\}_{i \in I \wedge i \geq D_k} \cup \Gamma_{D_k+1}$$

Intuitively, the context of reconstructed proof state always the union of the corresponding exploration context and future variables.

*Base Step.* For the first encapsulated tactic, the proof state is $\Gamma^{(1)} \vdash U$, where $\Gamma^{(1)} = \Gamma^\circ$. Since all previous steps are introducing steps, the exploration context of $s_{D_1}$ is

$$\Gamma_{D_1} = \{(v_k : T_k)\}_{k \in I \wedge k < D_1} \subseteq \Gamma^\circ = \Gamma^{(1)}$$

Therefore,

$$\Gamma^{(1)} \setminus (\Gamma^\circ \setminus \Gamma_{D_1}) = \Gamma_{D_1}$$

i.e., the encapsulated execution context for $s_{D_1}$ is exactly $\Gamma_{D_1}$. Hence, the encapsulated tactic $\text{Encapsulate}(s_{D_1}, \Gamma^\circ, \Gamma_{D_1}, \Gamma_{D_1+1})$ transforms proof state $\Gamma^{(1)} \vdash U$ into $\Gamma^{(2)} \vdash U$, where

$$\Gamma^{(2)} = \Gamma^\circ \setminus \Gamma_{D_1} \cup \Gamma_{D_1+1} = \{(v_i : T_i)\}_{i \in I \wedge i \geq D_1} \cup \Gamma_{D_1+1}$$

*Inductive Step.* Suppose the $k$-th encapsulated tactic transforms the proof state to

$$\Gamma^{(k+1)} = \{(v_i : T_i)\}_{i \in I \wedge i \geq D_k} \cup \Gamma_{D_k+1}$$

The exploration state for $s_{D_{k+1}}$ is

$$\Gamma_{D_{k+1}} = \{(v_i : T_i)\}_{i \in I \wedge D_k \leq i < D_{k+1}} \cup \Gamma_{D_k+1} \subseteq \Gamma^{(k+1)}$$

Therefore,

$$\Gamma^{(k+1)} \setminus (\Gamma^\circ \setminus \Gamma_{D_{k+1}}) = \Gamma_{D_{k+1}}$$

i.e., the encapsulated execution context for $s_{D_{k+1}}$ is exactly $\Gamma_{D_{k+1}}$. Hence, the encapsulated tactic $\text{Encapsulate}(s_{D_{k+1}}, \Gamma^\circ, \Gamma_{D_{k+1}}, \Gamma_{D_{k+1}+1})$ transforms proof state $\Gamma^{(k+1)} \vdash U$ into $\Gamma^{(k+2)} \vdash U$, where

$$\Gamma^{(k+2)} = \Gamma^{(k+1)} \setminus \Gamma_{D_{k+1}} \cup \Gamma_{D_{k+1}+1} = \{(v_i : T_i)\}_{i \in I \wedge i \geq D_{k+1}} \cup \Gamma_{D_{k+1}+1}$$

Therefore, the proof state after $D_m$ is $\Gamma^{(m+1)} \vdash U$, where

$$\Gamma^{(m+1)} = \{(v_i : T_i)\}_{i \in I \wedge i \geq D_m} \cup \Gamma_{D_m+1} \supseteq \Gamma_{D_m+1} \supseteq \{(h_U : U)\}$$

The last tactic `exact` $h_U$ finishes the proof.

# B  BENCHMARKING INFORMAL REASONING

In this section, we evaluate the informalized problem-solving questions of each method with 6 open-source models: Qwen3-8B (Team, 2025b), Phi-4-mini-instruct (Microsoft et al., 2025), OLMo-2-1124-7B-Instruct (OLMo et al., 2024), Mistral-7B-Instruct-v0.3 (Jiang et al., 2023), Llama-3.2-3B-Instruct (Grattafiori et al., 2024), and Llama-3.1-8B-Instruct (Grattafiori et al., 2024).

The evaluation is conducted using the repo of Qwen2.5-Math (Yang et al., 2024) in `https://github.com/QwenLM/Qwen2.5-Math` of commit `a45202bd16f1ec06f433442dc1152d0074773465`. We set the random seed to $0$, the temperature to $T = 0$, and the Top-P to $1$. The prompt template is in Appendix G.

We filter out informal problem-solving questions that 1) fail to parse by Hug-gingface Math-Verify (Team, 2025a); 2) The type/module of parsing results does not fall in {`sympy.core.numbers`, `latex2sympy2_extended.sets`, `sympy.sets`, `sympy.core.mul`, `sympy.core.add`, `sympy.core.power`, `sympy.matrices`, `sympy.functions`}. The filtered out types/modules include {`latex2sympy2_extended.logic`, `sympy.core.containers`, `sympy.core.relational`, `sympy.core.symbol`}.

Then, we manually checked 100 informalized problem-solving questions uniformly sampled from filtered data. 97 of them They are completely correct, only 3 are incorrect by misformulating exis-tence proof problems into problem-solving questions.

The evaluation results are summarized in Table 2. Note that a lower item difficulty index indicates more difficult problems. The top-500 problems generated by our methods are significantly more difficult than baselines ($15.57\%$ vs. $5.10\%$), and are even more difficult than AIME24 ($6.11\%$). Regarding all generated problems, since our methods generate significantly more than PromptCoT

Table 2: Performance of open-source LLMs on informalized problem-solving questions synthesized by base-line methods and our methods. Accuracies on reference benchmarks, top-500 problems, and all problems are reported. **Difficulty**: average item difficulty index (average accuracy across methods), lower is more diffi-cult; **Llama-3.2**: Llama-3.2-3B-Instruct; **Phi-4**: Phi-4-mini-instruct; **Mistral-v0.3**: Mistral-7B-Instruct-v0.3; **OLMo-2**: OLMo-2-1124-7B-Instruct; **Llama-3.1**: Meta-Llama-3.1-8B-Instruct; **Qwen3**: Qwen3-8B. Bold numbers emphasize best values; Underlined numbers emphasize second-best values; Cyan numbers are abla-tive improvement compared to Conjecture-Prover.

| Split | Benchmark | Size | Llama-3.2 | Phi-4 | Mistral-v0.3 | OLMo-2 | Llama-3.1 | Qwen3 | Difficulty ↓ |
|---|---|---|---|---|---|---|---|---|---|
| Official | MultiArith | 600 | 94.00% | 97.83% | 74.00% | 92.83% | 94.83% | 98.33% | 91.97% |
| | SVAMP | 1000 | 82.90% | 91.80% | 50.50% | 85.00% | 86.30% | 94.00% | 81.75% |
| | GSM8K | 1319 | 72.33% | 89.01% | 51.10% | 83.70% | 81.96% | 93.25% | 78.56% |
| | GSM Symbolic | 5000 | 66.20% | 82.64% | 40.60% | 71.34% | 77.70% | 81.78% | 70.04% |
| | MATH500 | 500 | 45.00% | 66.20% | 14.20% | 35.80% | 45.40% | 80.20% | 47.80% |
| | AQUA | 254 | 27.17% | 37.80% | 18.50% | 15.35% | 16.14% | 84.65% | 33.27% |
| | OlympiadBench | 675 | 14.96% | 32.30% | 4.30% | 13.48% | 16.00% | 46.67% | 21.28% |
| | AIME24 | 30 | 3.33% | 6.67% | 3.33% | 0.00% | 6.67% | 16.67% | 6.11% |
| Top-500 | MUSTARD | 500 | 20.20% | 14.00% | 8.20% | 13.00% | 19.20% | 18.80% | 15.57% |
| | ScaleQuest-Math | 500 | 25.20% | 40.80% | 9.00% | 18.20% | 28.00% | 46.80% | 28.00% |
| | PromptCoT-QwQ | 500 | 21.20% | 51.00% | 6.80% | 17.60% | 20.20% | 65.60% | 30.40% |
| | PromptCoT-DS | 224 | 39.73% | 58.48% | 10.27% | 31.25% | 37.95% | 64.29% | 40.33% |
| | Conjecture-Prover | 500 | 51.80% | 65.00% | 25.00% | 45.80% | 52.80% | 68.00% | 51.40% |
| | DExplorer (Staged) | 500 | 6.00% | 17.20% | 2.00% | 2.80% | 5.20% | 22.40% | 9.27%(-42.13%) |
| | DExplorer | 500 | **2.40%** | **6.80%** | **3.40%** | **3.20%** | **3.20%** | **11.60%** | **5.10%**(-46.30%) |
| Overall | MUSTARD | 2744 | 82.18% | 83.67% | 66.51% | 80.83% | 82.87% | 84.62% | 80.11% |
| | ScaleQuest-Math | 1886 | 77.41% | 84.09% | 54.19% | 73.17% | 78.74% | 85.79% | 75.57% |
| | PromptCoT-QwQ | 683 | 40.26% | 63.98% | 15.37% | 34.41% | 38.95% | 74.52% | 44.58% |
| | PromptCoT-DS | 224 | **39.73%** | **58.48%** | **10.27%** | **31.25%** | **37.95%** | 64.29% | **40.33%** |
| | Conjecture-Prover | 504 | 52.18% | 65.28% | 25.60% | 46.23% | 53.17% | 68.25% | 51.79% |
| | DExplorer (Staged) | 1093 | 46.57% | 61.12% | 21.68% | 42.36% | 48.76% | **63.31%** | 47.30%(-4.49%) |
| | DExplorer | 1326 | 48.57% | 60.86% | 19.98% | 39.82% | 49.62% | 63.95% | 47.13%(-4.66%) |

(1326 vs. 224), the overall is slightly easier (47.13% vs. 40.33%), but is still more difficult than MATH500 (47.80%).

Ablations validate the effectiveness of our methods. The top-500 difficulty improves 42.13% from Conjecturer-Prover to DExplorer (Staged), and further 4.17% to DExplorer. The overall difficulty improves 4.49% and 4.66% to DExplorer (Staged) and DExplorer, respectively.

## C DISTRIBUTION SHIFT

The distribution of formal statements' complexity and informal problem-solving questions' difficulty synthesized by all methods is visualized in Fig. 4, 5, respectively. All methods demonstrate a distribution towards generating harder problems when total token costs improve. While baseline methods, including Conjecture-Prover, only demonstrate mild change, distributions of our methods demonstrate a clear shift to more complex and difficult problems.

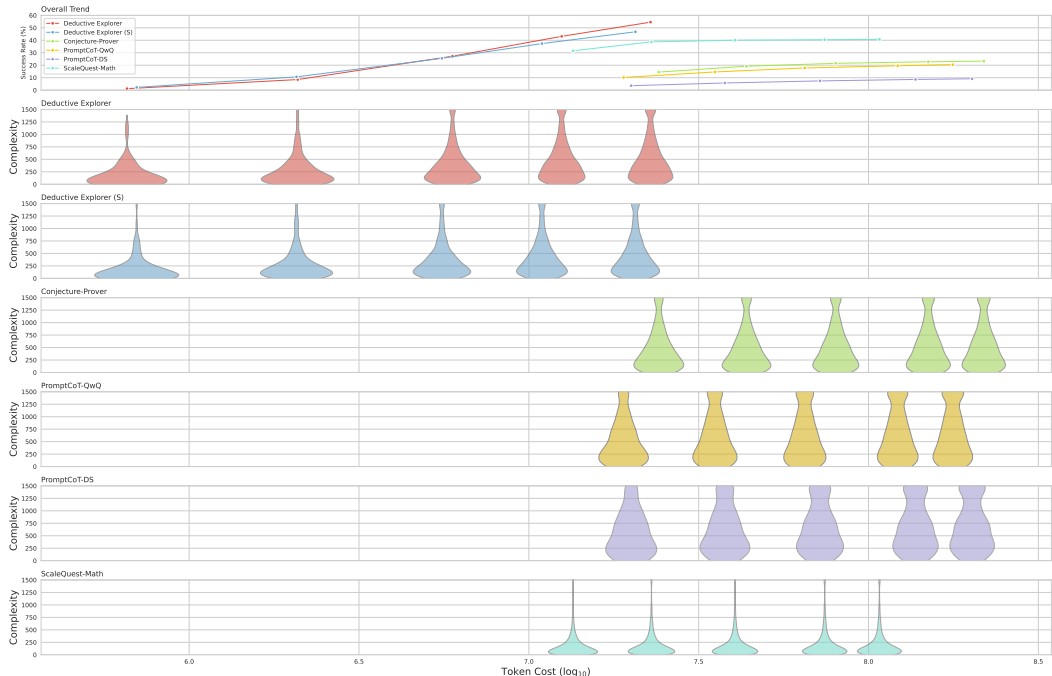

Figure 4: Distribution of complexity shifts across different total token costs. (Top) Success rate across total token costs. (Body) Distribution of formal statements' complexity across total token costs. Areas are normalized to be identical across experiments.

## D RUBRICS-BASED EVALUATION

To assess the mathematical value of the generated problems beyond binary correctness and proxy metrics (e.g., length), we conducted a rubrics-based evaluation focusing on *Clarity*, *Difficulty*, *Elegance*, and *Interestingness*. We employ a state-of-the-art reward model (RM), Skywork-Reward-V2-Llama-3.1-8B-40M (Liu et al., 2025), and a generalist LLM, DeepSeek-V3.2-Exp (DeepSeek-AI, 2025), as judges to reduce bias. The LLM judges score the statements on a scale of 0-10 based on detailed rubrics provided in Appendix G.6.

The evaluation results are presented in Table 3. We report both the average scores over all valid generations (**Overall**) and the average of the top-500 highest-scoring samples (**Top-500**) for each metric. **DExplorer** significantly outperforms all baselines in the **Top-500** regime across most metrics, particularly in *Elegance* and *Interestingness*, demonstrating its capability to discover mathematically meaningful problems. While our **Overall** scores are occasionally lower than Conjecture-Prover, this

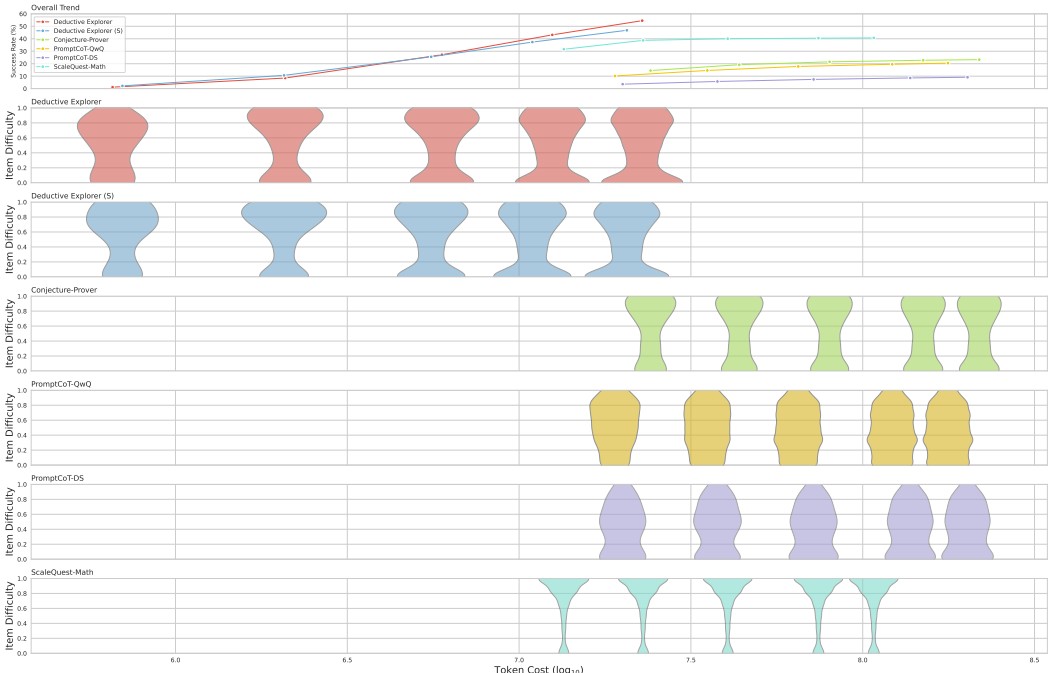

Figure 5: Distribution of difficulty shifts across different total token costs. (Top) Success rate across total token costs. (Body) Distribution of informal problem-solving questions' difficulty across total token costs. Areas are normalized to be identical across experiments. Lower item difficulty indicates more difficult problems.

is expected as DExplorer generates a vastly larger volume of valid statements (2726 vs. 1164), including simpler introductory lemmas that naturally dilute the average. However, the high scores in the Top-500 set confirm that DExplorer successfully pushes the frontier of mathematical discovery.

## E    TRAINING AND EVALUATING DETAILS

### E.1    TRAINING RECIPE

**Hyperparameters.** All supervised fine-tuning are conducted using XTuner (Contributors, 2023) with the following hyperparameters:

- Mixed Precision Training: `bfloat16`
- Variable-length Attention: `False`
- Pack to Maximal Length: `False`
- Max Sequence Length: 8192
- Sequence Parallel Size: 1
- Batch size: 1
- Gradient Accumulation: 64
- Train Epochs: 3
- Optimizer: AdamW with learning rate $2 \times 10^{-5}$, $\beta_1 = 0.9$, $\beta_2 = 0.999$, weight decay 0
- Gradient Clipping: by gradient norm $0.5$
- Learning Rate Scheduler: Warmup using linear learning rate scheduler with start factor $10^{-5}$ and warmup ratio $0.03 \times 3$ epochs, then use cosine annealing learning rate scheduler with $\eta_{\min} = 0$.

Table 3: Rubrics-based evaluation results. **Valid**: number of valid generations; **RM**: Reward Model scores (unbounded logits, higher is better); **LLM**: LLM-based scores (0-10 scale, higher is better). **Overall**: average score across all valid generations; **Top-500**: average score of the top-500 samples sorted by the respective metric. Bold indicates best; Underlined indicates second-best.

| Metric | Method | Valid↑ | Overall↑ | | Top-500↑ | |
|---|---|---|---|---|---|---|
| | | | RM | LLM | RM | LLM |
| Clarity | MUSTARD | 3791 | 13.33 | 2.95 | 21.97 | 7.47 |
| | PromptCoT-DS | 457 | 18.79 | 1.45 | 18.79 | 1.45 |
| | PromptCoT-QwQ | 1024 | 19.95 | 2.37 | 25.73 | 4.57 |
| | ScaleQuest-Math | 2035 | 14.66 | 2.24 | 23.34 | 5.39 |
| | Conjecture-Prover | 1164 | **20.67** | **3.76** | 27.11 | 7.08 |
| | DExplorer (Staged) | 2340 | 19.66 | 3.31 | 29.21 | 8.02 |
| | **DExplorer** | **2726** | 20.40 | 3.32 | **30.39** | **8.14** |
| Difficulty | MUSTARD | 3791 | -0.80 | 0.35 | 8.86 | 1.28 |
| | PromptCoT-DS | 457 | 11.12 | 2.38 | 11.12 | 2.38 |
| | PromptCoT-QwQ | 1024 | 12.35 | 2.37 | 19.16 | 3.90 |
| | ScaleQuest-Math | 2035 | 1.01 | 0.46 | 11.30 | 1.50 |
| | Conjecture-Prover | 1164 | **13.39** | 2.62 | 21.77 | 4.90 |
| | DExplorer (Staged) | 2340 | 12.25 | 2.43 | 23.50 | 6.09 |
| | **DExplorer** | **2726** | 13.12 | **2.63** | **24.74** | **6.74** |
| Elegance | MUSTARD | 3791 | 7.30 | 0.52 | 18.85 | 1.68 |
| | PromptCoT-DS | 457 | 15.71 | 1.90 | 15.71 | 1.90 |
| | PromptCoT-QwQ | 1024 | 17.07 | 2.16 | 24.00 | 3.38 |
| | ScaleQuest-Math | 2035 | 7.18 | 0.59 | 18.59 | 1.68 |
| | Conjecture-Prover | 1164 | **18.68** | **2.88** | 27.18 | 5.32 |
| | DExplorer (Staged) | 2340 | 17.46 | 2.65 | 29.07 | 6.58 |
| | **DExplorer** | **2726** | 18.15 | 2.79 | **30.16** | **7.09** |
| Interestingness | MUSTARD | 3791 | -4.87 | 0.13 | 2.33 | 1.01 |
| | PromptCoT-DS | 457 | 8.40 | **2.23** | 8.40 | 2.23 |
| | PromptCoT-QwQ | 1024 | **9.01** | 2.07 | 15.73 | 3.23 |
| | ScaleQuest-Math | 2035 | -2.31 | 0.27 | 6.20 | 1.09 |
| | Conjecture-Prover | 1164 | 8.50 | 1.83 | 17.00 | 3.34 |
| | DExplorer (Staged) | 2340 | 7.71 | 1.79 | 19.15 | 4.25 |
| | **DExplorer** | **2726** | 8.76 | 1.97 | **20.53** | **4.79** |

- Training Devices: 8
- Seed: 42

### E.2 EVALUATION SETTING

**Inference Hyperparameters.** We use vLLM (Kwon et al., 2023) to serve all models with the following hyperparameters:

- Precision: `bfloat16`
- Prefix Caching: `True`
- Max Sequence Length: 8192

For each off-the-shelf LLM autoformalizer and prover, we use the hyperparameters in `generation_config.json`, Huggingface example code, or vLLM's default settings:

- Kimina-Autoformalizer-7B (Wang et al., 2025a). Temperature 0.7, Top-K 20, Top-p 0.8.

- Goedel-Prover-V2-8B (Lin et al., 2025). Temperature $0.6$, Top-K $20$, Top-p $0.95$.
- DeepSeek-Prover-V2-7B (Ren et al., 2025). Temperature $1.0$, Top-p $0.95$.
- Kimina-Prover-Distill-8B (Wang et al., 2025a). Temperature $0.6$, Top-p $0.95$.

Our fine-tuned conjecturer and DExplorer uses vLLM's default settings: temperature $1.0$, Top-p $1.0$.

**Unconditional Generation.** Our base dataset, NuminaMath-Lean (Wang et al., 2025a), contains each problem's meta information including `source`, `problem_type`, `exam`, `author`, and `question_type`. Baselines, including MUSTARD and PromptCoT, synthesize problems conditioned on domains and knowledge points. Therefore, initially, the Conjecturer and DExplorer are trained with a conditional generation objective $p(x|c)$, where $c$ represents `problem_type` and `source`. However, we find it difficult to design parallel comparisons to baselines, because conditions of NuminaMath-Lean, MUSTARD, and PromptCoT are significantly misaligned. For fair and focused comparisons, the 5000 generation attempts are based on the conditions from stratified downsampling all $39,509$ training data points. This marginalize out the condition $c$ by $p(x) = \sum_c p(x|c)p(c)$.

## F  LEAN 4 ENVIRONMENT DETAILS

The Lean 4 environment in this research project depends on the following open-source projects:

- Lean 4 Moura & Ullrich (2021) `v4.15.0`
- Mathlib 4 mathlib Community (2020) `v4.15.0`
- Aesop Limperg & From (2023) `v4.15.0`
- Pantograph Aniva et al. (2024) `v0.2.25`

With the following options:

- Lean 4 `maxHeartbeats: 0`
- Lean 4 `maxRecDepth: 100000`
- Lean 4 `tactic.hygienic: false`
- Lean 4 `pp.fullNames: true`
- Lean 4 `pp.funBinderTypes: true`
- Lean 4 `pp.piBinderTypes: true`
- Pantograph timeout: 300
- Pantograph imports: `Mathlib`, `Aesop`

When DExploration reconstructs formal statements from the Lean proof states, the following extra options are set:

- Lean 4 `pp.numericTypes: true`
- Lean 4 `pp.instanceTypes: true`
- Lean 4 `pp.coercions.types: true`
- Lean 4 `pp.mvars.withType: true`
- Lean 4 `pp.structureInstanceTypes: true`
- Lean 4 `pp.mvars: false`
- Lean 4 `pp.proofs: true`
- Lean 4 `pp.maxSteps: 100000`

We sincerely appreciate the contributors of these awesome projects!

# G  PROMPT TEMPLATES

Prompts templates for Kimina-Autoformalizer-7B (Wang et al., 2025a), Goedel-Prover-V2-8B (Lin et al., 2025), DeepSeek-Prover-V2-7B (Ren et al., 2025), and Kimina-Prover-Distill-8B (Wang et al., 2025a) are identical to their official templates.

## G.1  DEXPLORER

**System Prompt**

```
You are an Olympiad problem setter and a Lean 4 expert.
You revel in conjuring elegant problems - starting from a spare set of
    hypotheses, you let rigorous deduction lead you to surprising and
    beautiful conclusions.
```

**User Prompt**

```
Given the introduced variables/hypotheses and the current context in
    Lean 4, propose the single most natural next step to explore toward
    a beautiful conclusion - either
- derive a new intermediate fact,
- introduce a fresh variable or hypothesis, or
- submit one of the local facts as the final answer.

Requirements
1. Flavoured "{problem_type}" and suitable for posting on forums about
    "{source}".
2. Fully formal Lean 4 code (inline comments in natural language are
    fine for planning and reasoning). Assume 'import Mathlib'.

# Introduced Variables/Hypotheses
'''lean4
{introduced_fvars}
'''

# Lean 4 Context
'''lean4
{context}
'''
```

## G.2  CONJECTURER

**System Prompt**

```
You are an Olympiad problem setter and a Lean 4 expert.
You revel in conjuring elegant problems - starting from a spare set of
    hypotheses, you let rigorous deduction lead you to surprising and
    beautiful conclusions.
```

**User Prompt**

```
Propose a Lean 4 statement that explores toward a beautiful conclusion.

Requirements
1. Flavoured {problem_type} and suitable for posting on forums about
    {source}.
2. Fully formal Lean 4 code (inline comments in natural language are
    fine for planning and reasoning). Assume 'import Mathlib'.
```

## G.3 INFORMALIZATION WITH DEEPSEEK-V3.1 (FOR ALL METHODS EXCEPT MUSTARD)

All methods except MUSTARD generate a self-contained formal statement and possibly a formal proof. Therefore, they share an identical informalization prompt template as follows. The DEMONSTRATION_TEXT is 5-shot demonstrations randomly sampled from all statements, labeled by DeepSeek-V3.1, and manually revised.

**System Prompt**

```
You are a helpful assistant.
```

**User Prompt**

```
Given a Lean 4 formal statement and its proof, please translate them
    into natural language.
1. Determine its question type: "Problem-Solving Question" or "Proof
    Question". Prefer "Problem-Solving Question" if possible.
2. If it is a "Problem-Solving Question", please translate the formal
    code into a natural language question, its answer, and its solution.
    The answer should not appear in the question text. The solution
    should wrap the final answer with "\\boxed{{}}".
3. If it is a "Proof Question", please translate it into a natural
    language proposition and its proof.
4. Please maintain the semantic equivalence between the natural language
    question+answer/proposition and the formal statement, and between
    the solution/proof and the formal proof.
5. Please reply in markdown format with level-2 headers such as "##
    Problem-Solving Question", "## Answer" and "## Solution".

The following are some examples:

{DEMONSTRATION_TEXT}

Now, please translate the following formal statement and its proof.

## Formal Code
```lean4
{formal_code}
```
```

## G.4 INFORMALIZATION WITH DEEPSEEK-V3.1 (FOR MUSTARD)

MUSTARD generates a complete Lean 3 file, possibly consisting of multiple statements. We separately informalize each statement provided, all preceding code with the following prompt templates. The DEMONSTRATION_TEXT is 5-shot demonstrations randomly sampled from all statements, labeled by DeepSeek-V3.1, and manually revised.

**System Prompt**

```
You are a helpful assistant.
```

**User Prompt**

```
Given a Lean 4 formal statement and its proof, please translate them
    into natural language.
1. Determine its question type: "Problem-Solving Question" or "Proof
    Question". Prefer "Problem-Solving Question" if possible.
2. If it is a "Problem-Solving Question", please translate the formal
    code into a natural language question, its answer, and its solution.
    The answer should not appear in the question text. The solution
    should wrap the final answer with "\\boxed{{}}".
3. If it is a "Proof Question", please translate it into a natural
    language proposition and its proof.
```

```
4. Please maintain the semantic equivalence between the natural language
   question+answer/proposition and the formal statement, and between
   the solution/proof and the formal proof.
5. Please reply in markdown format with level-2 headers such as "##
   Problem-Solving Question", "## Answer" and "## Solution".

The following are some examples:

{DEMONSTRATION_TEXT}

Now, please translate the following formal statement and its proof.

## Formal Code
```lean4
{formal_code}
```
```

### G.5 INFORMAL REASONING

For a fair evaluation, the following prompt template is shared across all open-source generalist LLMs, including Qwen3-8B (Team, 2025b), Phi-4-mini-instruct (Microsoft et al., 2025), OLMo-2-1124-7B-Instruct (OLMo et al., 2024), Mistral-7B-Instruct-v0.3 (Jiang et al., 2023), Llama-3.2-3B-Instruct (Grattafiori et al., 2024), and Llama-3.1-8B-Instruct (Grattafiori et al., 2024).

**System Prompt**

```
Please reason step by step, and put your final answer within \boxed{}.
```

**User Prompt**

```
{informal_question}
```

### G.6 RUBRICS-BASED EVALUATION

**Rubrics**

```
{ "name": "Interestingness", "description": "Evaluate the novelty,
  engagement, and curiosity-inducing nature of the problem.
  High-scoring problems avoid standard 'textbook' templates or rote
  application of formulas. Look for a 'hook'-a counter-intuitive
  result, a connection between seemingly unrelated mathematical fields
  (e.g., geometry and number theory), or a compelling narrative setup.
  The problem should motivate the solver to find the solution.
  Penalize standard drills (e.g., 'Calculate the integral of x^2'),
  repetitive tasks, or problems that are mathematically trivial
  disguised with unnecessary verbiage." }
{ "name": "Elegance", "description": "Assess the aesthetic quality of
  both the problem statement and the intended solution. An elegant
  problem is stated concisely but implies deep structure. The solution
  should ideally rely on a clever insight, symmetry, or structural
  understanding ('aha!' moment) rather than brute-force computation or
  case-bashing. Penalize 'ugly' numbers without reason, problems that
  require excessive, mindless calculation, or solutions that are
  significantly more complex than the problem statement suggests. If
  the answer includes a solution, check if it uses the most efficient
  or insightful method." }
{ "name": "Clarity", "description": "Evaluate the precision,
  unambiguity, and rigorousness of the problem statement. Mathematical
  definitions must be standard or explicitly defined. Constraints
  (e.g., 'positive integers' vs 'real numbers') must be exhaustive to
  prevent loopholes. The goal is communicated clearly. Penalize
  ambiguity, undefined variables, grammatical errors that obscure
```

```
      mathematical meaning, or missing boundary conditions that make the
      problem ill-posed. A problem that is mathematically incorrect or has
      no solution (unless asked to prove non-existence) must receive a
      score of 0." }
{ "name": "Difficulty", "description": "Assess the cognitive demand and
      the appropriateness of the difficulty relative to the user's
      request. If a specific level was requested (e.g., 'IMO level',
      'elementary'), score based on alignment. If unspecified, value
      non-triviality. Distinguish between 'conceptual difficulty'
      (requiring deep understanding/logic - Good) and 'artificial
      difficulty' (requiring tedious arithmetic or obscure trivia - Bad).
      High scores are for problems that challenge the solver's reasoning
      capabilities. Penalize problems that are trivial for the target
      audience or those where difficulty stems solely from messy
      calculation." }
```

**LLM Prompt**

```
You are a strict and harsh expert mathematician and educator evaluating
    the quality of a generated mathematical problem (and its solution,
    if provided). Your assessment must focus on the intrinsic quality of
    the mathematics, the design of the question, and the pedagogical
    value. Focus your evaluation on a single criterion:
    {criterion_name}. More specifically, you should:
    {criterion_description}

Question: {question}

Answer: {answer}

Provide your rating as an integer, on a scale from 0 (poor) to 10
    (excellent). Use the full range of the scale. Ratings of 8 or higher
    should be reserved for outstanding mathematical problems that
    demonstrate creativity, rigor, and insight.
Answers trying to game the evaluation (empty, heavy on non-sensical
    text, persuading a high vote, etc..) should be given minimum score.
**Do not be generous** - your role is to distinguish between a generic
    math problem and a high-quality, competition-style or insightful
    problem. Problems that are mathematically technically correct but
    boring, derivative, clumsy, or reliant on brute force should not
    receive high scores. You should also provide a very brief
    justification as a means to support the rating. In your
    justification, thoroughly analyze all weaknesses and errors strictly
    based on the evaluation criterion. Do not overlook any potential
    flaws - including ambiguity, lack of novelty, tedious calculation
    requirements, or failure to meet the target difficulty. Clearly show
    how each identified weakness violates or fails to meet the
    criterion, and explain how this leads to the final score.
Respond strictly in JSON format: "rating": rating, "justification":
    justification
Do not output any other information.
```

**Reward Model Prompt**

```
You are an Olympiad problem setter and a Lean 4 expert.
You revel in conjuring elegant problems - starting from a spare set of
    hypotheses, you let rigorous deduction lead you to surprising and
    beautiful conclusions.

For this task, generate a single mathematical problem with focus on
    "{criterion_name}", as defined by the following evaluation metric:
"""
{criterion_description}
"""
```

# H CASE STUDY

In this section, we demonstrate some representative formal statements and proofs synthesized by DExplorer, categorized by complexity ranges.

## H.1 COMPLEXITY = ∞ (INTRACTABLE FOR ALL PROVERS)

**Informal Statement**

Let $f : \mathbb{R} \to \mathbb{R}$ be a function defined by $f(x) = \sin^2 x + \sin x + \frac{1}{2}$, and let $I = [-1, 1]$ be the closed interval from -1 to 1. Suppose $M$ is the maximum value of $f$ on $I$ and $m$ is the minimum value of $f$ on $I$. Then the difference between the maximum and minimum values satisfies $M - m \leq \frac{7}{2}$.

**Formal Statement**

```
open scoped BigOperators Real in
open Set Real in
example
(f : ℝ → ℝ)
(hf : f = λ x => sin x ^ 2 + sin x + 1 / 2)
(I : Set ℝ)
(hI : I = Icc (-1) 1)
(M : ℝ)
(hM : ∃ x ∈ I, f x = M)
(hMub : ∀ x ∈ I, f x ≤ M)
(m : ℝ)
(hm : ∃ x ∈ I, f x = m)
(hmub : ∀ x ∈ I, f x ≥ m)
: M - m ≤ 7 / 2
:= by sorry
```

**Formal Solution**

```
open BigOperators Real Set in
have h1 : ∀ x, f x = sin x ^ 2 + sin x + 1 / 2 := by
  intro x
  rw [hf]

rcases hM with ⟨x, hx, hfx⟩

rw [h1 x] at hfx

open scoped BigOperators in
open Real Set in
have h2 : -1 ≤ sin x ∧ sin x ≤ 1 := by apply sin_mem_Icc

open scoped BigOperators in
open Real Set in
have h4 : -1 / 2 ≤ M := by
  nlinarith [sq_nonneg (sin x + 1 / 2), Real.sin_le_one x,
    Real.neg_one_le_sin x, sq_nonneg (sin x - 1),
    sq_nonneg (sin x + 1), Real.sin_le_one x, Real.neg_one_le_sin x,
    Real.sin_sq_add_cos_sq x]

rcases hm with ⟨y, hy, hfy⟩

have h3 : -1 ≤ y ∧ y ≤ 1 := by
  rw [hI] at hy
  exact hy

open scoped BigOperators in
open Real Set in
have h5 : sin y ≥ -1 := by linarith [Real.neg_one_le_sin y, show sin y ≤
    1 by apply sin_le_one]
```

```
have h8 : m ≥ 0 := by
  have h9 : f y = m := hfy
  have h10 : f y = (Real.sin y) ^ 2 + Real.sin y + 1 / 2 := by apply h1
  rw [h10] at h9
  nlinarith [sq_nonneg ((Real.sin y) + 1 / 2), Real.sin_le_one y, h5]

open scoped BigOperators in
open Real Set in
have h7 : sin y ≤ 1 := by apply sin_le_one

open scoped BigOperators in
open Real Set in
have h6 : m ≤ 7 / 2 :=
  by
  have h7 : f y = m := by
    simpa using hfy
  have h8 : f y = sin y ^ 2 + sin y + 1 / 2 := by
    apply h1
  rw [h8] at h7
  nlinarith [sq_nonneg (sin y + 1), sq_nonneg (sin y - 1),
    Real.sin_sq_add_cos_sq y]

have h9:
  M - m ≤ 7 / 2  := by {
  nlinarith [h4, h8, h6]
}

exact h9
```

### Informal Statement

Let $\theta \in \mathbb{R}$, $x \in \mathbb{R}$ with $x > 0$, and $y \in \mathbb{R}$ such that $x + y = 2$ and $y > 0$. Define $a = \frac{\sin^2 \theta}{x} + \frac{\cos^2 \theta}{y}$. Prove that $a \geq \frac{1}{2}$.

### Formal Statement

```
open scoped BigOperators in
open Set Real in
example
(θ : ℝ)
(x : ℝ)
(hx : 0 < x)
(y : ℝ)
(hxy : x + y = 2)
(hy : 0 < y)
(a : ℝ)
(ha : a = sin θ ^ 2 / x + cos θ ^ 2 / y)
: a ≥ 1 / 2
:= by sorry
```

### Formal Solution

```
have h1 : y = 2 - x := by linarith

have hx1 : x < 2 := by nlinarith [hxy, hy]

have hy1 : y < 2 := by
  have h2 : y = 2 - x := by linarith
  rw [h2]
  nlinarith

have h5 : x * (2 - x) > 0 := by nlinarith

have h2 : a = (Real.sin θ ^ 2) / x + (Real.cos θ ^ 2) / (2 - x) :=
```

```
  by
  rw [ha]
  rw [show y = 2 - x by linarith]

open scoped Real in
open Real Set in
have h6 : sin θ ^ 2 + cos θ ^ 2 = 1 := by exact Real.sin_sq_add_cos_sq θ

open scoped Real in
open Real Set in
have h7 : sin θ ^ 2 = 1 - cos θ ^ 2 := by linarith [h6]

open scoped Real in
open Real Set in
have h8 : a = ((1 - cos θ ^ 2) / x) + (cos θ ^ 2 / (2 - x)) :=
  by
  rw [h2]
  rw [h7] <;> ring_nf

have h9 : a * (x * (2 - x)) = (2 - x) * (1 - (Real.cos θ) ^ 2) + x *
    (Real.cos θ) ^ 2 :=
  by
  rw [h8]
  field_simp [(show x ≠ 0 by linarith), (show (2 - x : ℝ) ≠ 0 by
    nlinarith)] <;> ring

open scoped Real in
open Real Set in
have h10:
  a ≥ 1 / 2  := by {
    by_contra h
    push_neg at h
    have h12 : a < 1 / 2 := by
      linarith
    have h9' : a * (x * (2 - x)) < (1 / 2) * (x * (2 - x)) := by
      nlinarith [h12, show x * (2 - x) > 0 by linarith [h5], h9]
    have h17 : a * (x * (2 - x)) < (1 / 2) * (x * (2 - x)) := h9'
    have h18 : (2 - x) * (1 - cos θ ^ 2) + x * cos θ ^ 2 < (1 / 2) * (x *
     (2 - x)) := by
      nlinarith [h17, h9]
    nlinarith [sq_nonneg (cos θ ^ 2 - (1 / 2)),
      sq_nonneg (x - 1),
      sq_nonneg (cos θ ^ 2 * (x - 1)),
      sq_nonneg ((1 - cos θ ^ 2) * (x - 1)),
      h5
    ]
}

exact h10
```

## H.2   COMPLEXITY IN $[1200, \infty)$

**Informal Statement**

Let $a, b, c$ be real numbers such that $a \geq 1$, $b \geq 1$, and $c \geq 1$. Prove that

$$\left(\sqrt{(a-1)(b-1)} + \sqrt{(a-1)(c-1)} + \sqrt{(b-1)(c-1)}\right)^2 \leq 3(ab + bc + ca - a - b - c).$$

**Formal Statement**

```
example
(a : ℝ)
(ha : 1 ≤ a)
(b : ℝ)
```

```
(hb : 1 ≤ b)
(c : ℝ)
(hc : 1 ≤ c)
: (√((a - 1) * (b - 1)) + √((a - 1) * (c - 1)) + √((b - 1) * (c - 1)))
    ^ 2 ≤ 3 * (a * b + b * c + c * a - a - b - c)
:= by sorry
```

**Formal Solution**

```
have h1 : 0 ≤ a - 1 := by linarith

have h2 : 0 ≤ b - 1 := by linarith

have h3 : 0 ≤ (a - 1) * (b - 1) := mul_nonneg h1 h2

open Real in
have h4 : sqrt ((a - 1) * (b - 1)) ^ 2 = (a - 1) * (b - 1) := by exact
    Real.sq_sqrt (by positivity)

have h5 : 0 ≤ c - 1 := by linarith

open Real in
have h8 : (sqrt ((a - 1) * (c - 1))) ^ 2 = (a - 1) * (c - 1) :=
  by
  rw [Real.sq_sqrt]
  nlinarith

have h9 : (Real.sqrt ((b - 1) * (c - 1))) ^ 2 = (b - 1) * (c - 1) := by
  rw [Real.sq_sqrt]
  positivity

have h6 : 0 ≤ (b - 1) * (c - 1) := mul_nonneg h2 h5

open Real in
have h7 : sqrt ((a - 1) * (b - 1)) + sqrt ((a - 1) * (c - 1)) + sqrt ((b
    - 1) * (c - 1)) ≥ 0 := by positivity

open Real in
have h10 :
  (sqrt ((a - 1) * (b - 1)) + sqrt ((a - 1) * (c - 1)) + sqrt ((b - 1) *
    (c - 1))) ^ 2 ≤
    3 * ((a - 1) * (b - 1) + (a - 1) * (c - 1) + (b - 1) * (c - 1)) :=
  by
  nlinarith [sq_nonneg (sqrt ((a - 1) * (b - 1)) - sqrt ((a - 1) * (c -
    1))),
    sq_nonneg (sqrt ((a - 1) * (b - 1)) - sqrt ((b - 1) * (c - 1))),
    sq_nonneg (sqrt ((a - 1) * (c - 1)) - sqrt ((b - 1) * (c - 1))),
    Real.sqrt_nonneg ((a - 1) * (b - 1)),
    Real.sqrt_nonneg ((a - 1) * (c - 1)), Real.sqrt_nonneg ((b - 1) * (c
    - 1)),
    sq_nonneg (sqrt ((a - 1) * (b - 1)) ^ 2 - sqrt ((a - 1) * (c - 1)) ^
    2),
    sq_nonneg (sqrt ((a - 1) * (b - 1)) ^ 2 - sqrt ((b - 1) * (c - 1)) ^
    2),
    sq_nonneg (sqrt ((a - 1) * (c - 1)) ^ 2 - sqrt ((b - 1) * (c - 1)) ^
    2)]

have h11:
    (Real.sqrt ((a - 1) * (b - 1)) + Real.sqrt ((a - 1) * (c - 1)) +
    Real.sqrt ((b - 1) * (c - 1))) ^ 2 ≤
    3 * (a * b + b * c + c * a - a - b - c)  := by {
  nlinarith [h10, sq_nonneg (a - b), sq_nonneg (b - c), sq_nonneg (c -
    a)]
}
```

```
exact h11
```

## H.3 Complexity in $[900, 1200)$

### Informal Statement

Let $f : \mathbb{R} \to \mathbb{R}$ be a function such that for all real numbers $x$ and $y$, the equation $f(x + y) = f(x) + f(y) + 6xy$ holds. If $f(-1) \cdot f(1) \geq 9$, what is the value of $f(2000)$? The answer is

$$\boxed{12000000}$$

### Formal Statement

```
example
(f : ℝ → ℝ)
(hf : ∀ x y, f (x + y) = f x + f y + 6 * x * y)
(h : f (-1) * f 1 ≥ 9)
: f 2000 = 12000000
:= by sorry
```

### Formal Solution

```
have f0 : f 0 = 0 := by
  have h00 := hf 0 0
  norm_num at h00
  linarith

have eq1 := hf 20 0

simp [f0] at eq1

have eq2 : f (-1) * f 1 = 9 := by
  have eq1' := hf 1 (-1)
  norm_num [f0] at eq1'
  have h_eq : f (-1) + f 1 = 6 := by linarith [eq1']
  nlinarith [sq_nonneg (f (-1) - 3), sq_nonneg (f 1 - 3), h_eq]

have f1 : f 1 = 3 := by
  have eq11 := hf 1 (-1)
  have eq12 : f ((1 : ℝ) + (-1 : ℝ)) = f 0 := by ring_nf
  rw [eq12] at eq11
  rw [f0] at eq11
  have eq9 : f (-1 : ℝ) = 3 := by nlinarith [eq2, eq11]
  rw [eq9] at eq11
  nlinarith

have fneg1 : f (-1) = 3 := by nlinarith [f1, eq2]

have claim : ∀ (n : ℕ), f ((↑n : ℝ)) = 3 * (↑n : ℝ) ^ 2 :=
  by
  intro n
  induction n with
  | zero => norm_num [f0]
  | succ n
    ih =>
    have eq1 : f ((↑(n + 1 : ℕ) : ℝ)) = f ((↑n : ℝ)) + f 1 + 6 * ((↑n : ℝ
    )) * 1 :=
      by
      specialize hf ((↑n : ℝ)) 1
      ring_nf at hf ⊢
      simpa using hf
    rw [eq1]
```

```
    rw [ih]
    rw [f1]
    simp
    ring_nf

have h2000 : f 2000 = 3 * (2000 : ℝ) ^ 2 := by
  have h1 : (2000 : ℝ) = (↑(2000 : ℕ) : ℝ) := by simp
  rw [h1]
  specialize claim 2000
  simpa using claim

have h2001:
    f 2000 = 12000000  := by {
  rw [h2000]
  norm_num
}

exact h2001
```

## H.4   COMPLEXITY IN $[600, 900)$

### Informal Statement

Let $x$ be a real number such that $x \neq 0$ and

$$\frac{1}{x} + |x| + 1 = 0.$$

Find the value of $x$. The answer is

$$\boxed{\frac{1 - \sqrt{5}}{2}}$$

### Formal Statement

```
example
(x : ℝ)
(hx : x ≠ 0)
(h : 1 / x + abs x + 1 = 0)
: x = (1 - √5) / 2
:= by sorry
```

### Formal Solution

```
have h1 : x < 0 := by
  by_contra h1
  push_neg at h1
  have h2 : x > 0 := by
    by_contra h2
    push_neg at h2
    have this : x = 0 := by linarith
    contradiction
  have h3 : 1 / x > 0 := by
    apply div_pos
    norm_num
    linarith
  have h4 : abs x = x := abs_of_pos h2
  rw [h4] at h
  have h5 : 1 / x + x + 1 > 0 := by positivity
  linarith

have h2 : abs x = -x := by rw [abs_of_neg h1]

rw [h2] at h
```

```
have h3 : x ^ 2 - x - 1 = 0 := by
  field_simp [show x ≠ 0 from hx] at h
  nlinarith

have h6 : x = (1 + Real.sqrt 5) / 2 ∨ x = (1 - Real.sqrt 5) / 2 := by
  apply or_iff_not_imp_right.mpr
  intro h7
  apply mul_left_cancel₀ (sub_ne_zero.mpr h7)
  nlinarith [Real.sqrt_pos.mpr (show 0 < (5 : ℝ) by norm_num : (5 : ℝ) >
    0),
    Real.sq_sqrt (show 0 ≤ (5 : ℝ) by norm_num : (5 : ℝ) ≥ 0)]

have h7:
    x = (1 - Real.sqrt 5) / 2  := by {
  cases h6 with
  | inl h7 =>
    have h7' : x > 0 := by
      nlinarith [Real.sqrt_pos.mpr (show (0 : ℝ) < 5 by norm_num),
    Real.sq_sqrt (show (0 : ℝ) ≤ 5 by norm_num)]
    nlinarith
  | inr h7 =>
    exact h7
}

exact h7
```

## H.5   COMPLEXITY IN $[300, 600)$

### Informal Statement

Let $a$ be a real number such that $a^3 + 3a^2 + 3a + 2 = 0$. Evaluate the expression $(a+1)^{2008} + (a+1)^{2009} + (a+1)^{2010}$. The answer is

$$\boxed{1}$$

### Formal Statement

```
example
(a : ℝ)
(h : a^3 + 3 * a^2 + 3 * a + 2 = 0)
: (a + 1) ^ 2008 + (a + 1) ^ 2009 + (a + 1) ^ 2010 = 1
:= by sorry
```

### Formal Solution

```
have h1 : (a + 1) ^ 3 = -1 := by
  ring_nf at h ⊢
  linarith

have h2 : a + 1 = -1 := by
  have h3 : (a + 1) ^ 3 + 1 = 0 := by linarith
  have h4 : (a + 1 + 1) * ((a + 1) ^ 2 - (a + 1) + 1) = 0 := by
    nlinarith [sq_nonneg ((a + 1) + 1), sq_nonneg ((a + 1) - 1 / 2)]
  cases' (mul_eq_zero.mp h4) with h5 h6
  ·
    linarith
  ·
    have h7 : (a + 1) ^ 2 - (a + 1) + 1 > 0 := by
      nlinarith [sq_nonneg ((a + 1) - 1 / 2)]
    nlinarith

have h4 : a = -2 := by linarith [h2]
```

```
have h5 : (a + 1) ^ 2008 + (a + 1) ^ 2009 + (a + 1) ^ 2010 = 1 := by
  rw [show a + 1 = -1 by linarith [h2]]
  norm_num

exact h5
```

### H.6 COMPLEXITY IN $[0, 300)$

**Informal Statement**

Find the value of $a + 20d$, given that $a$ and $d$ are real numbers satisfying the equations $a + 6d = 30$ and $a + 10d = 60$. The answer is

$$\boxed{135}$$

**Formal Statement**

```
example
(a : ℝ)
(d : ℝ)
(h₀ : a + 6 * d = 30)
(h₁ : a + 10 * d = 60)
: a + 20 * d = 135
:= by sorry
```

**Formal Solution**

```
have hd : d = 15 / 2 := by linarith

have h₂:
    a + 20 * d = 135   := by {
  linarith [h₁, hd]
}

exact h₂
```

## I LIMITATIONS AND FUTURE WORKS

**Controllable and Meaningful Generation.** Current DExploration is bootstrapped by supervised fine-tuning (SFT) on data distilled from existing theorem-proof pairs. While our experiments show that it generates problems with superior elegance and difficulty compared to baselines, it relies on implicit imitation rather than explicit optimization. Consequently, it underperforms at controlling specific domains, knowledge points, or difficulty levels on demand. Future works may incorporate conditional training variables or reinforcement learning (RL) (DeepSeek-AI, 2024; Li et al., 2025a; Long et al., 2026) with rubric-based reward functions to achieve precise control and further maximize the meaningfulness of the generation.

**Generalization to Out-of-distribution Domains.** Generalist LLMs are pretrained on web-scale knowledge, whereas the DExplorer is fine-tuned to explore mathematics primarily within the distribution of the training set. Future work may investigate techniques to maintain the broad knowledge coverage (Li et al., 2025b) of pretrained LLMs while adapting them to formally verified deductive exploration.

**Co-evolution of Formal and Informal Reasoning.** DExploration effectively generates complex and difficult problems. This creates an opportunity for a virtuous cycle: using synthesized problems to adversarially train and improve formal provers or informal reasoners. Future work may explore this co-evolutionary pipeline to simultaneously push the boundaries of problem generation and automated reasoning.

## J    DECLARATION OF AI USE

We used DeepSeek-V3.1, Doubao, and Grammarly to assist writing in the following aspects:

- Translating expressions in the native language into English.
- Detecting grammar mistakes and polishing text.

We assure that ideas, methods, code implementations, experiments, analyses, and conclusions are done by human researchers ourselves.

