# OpenReview forum: "Let's Explore Step by Step: Generating Provable Formal Statements with Deductive Exploration"
_ICLR.cc/2026/Conference — ICLR 2026 Poster_

### Official Review · Reviewer_ZhGX · 2025-10-26

**Soundness:** 3
**Presentation:** 3
**Contribution:** 3
**Rating:** 6
**Confidence:** 4

**Summary:**

I think the authors did a genuinely impressive job with this paper. Let’s Explore Step by Step tackles one of the hardest open challenges in AI4Math: how to synthesize new, provably valid mathematical problems without relying on fragile external provers or autoformalizers. The proposed framework, DExploration, is both elegant and well-motivated. Instead of one-shot problem generation, the model explores mathematics step by step ie introducing variables, deducing facts, and finally submitting a conclusion — all grounded within the Lean 4 proof assistant.

This design ensures whole-process verifiability: every generated problem is backed by a valid proof trace, eliminating the need for an external verifier. I especially like how the authors handle data scarcity through Exploratory Transformation, which rewrites existing proofs into “exploration trajectories” via dependency graphs and topological ordering. It’s a clever and principled way to bootstrap training data from existing Lean corpora.

**Strengths:**

I think the authors have really nailed something important here with the core idea of DExploration. Instead of trying to generate entire mathematical statements in one shot, they've broken it down into a natural exploration process—introducing variables, deducing facts, and building up to conclusions step-by-step. This feels much more aligned with how mathematicians actually work, and the fact that every single step is verified in Lean 4 is brilliant. It completely sidesteps the validity issues that plague other approaches. The framework doesn't just generate statements; it simultaneously constructs their proofs, which means no need for external provers that might fail on harder problems. That's a genuinely clever solution to what the authors call the "expressiveness-validity-complexity trilemma," and I find it both theoretically elegant and practically useful.

The experimental work is really thorough and convincing. Good job on testing across so many different dimensions; they don't just show their method works, they show it works *better* in multiple ways simultaneously. The 83% reduction in token costs alone is impressive, but combine that with higher success rates (40.70% to 54.52%), better complexity distributions, and harder problems that actually stump current state-of-the-art provers? That's strong evidence this approach is onto something. I particularly appreciate how they've evaluated both formal and informal reasoning, showing the generated problems are harder than AIME24 when translated to natural language. The ablation studies are well-designed too they clearly demonstrate that both the exploration framework and the exploratory transformation pipeline are pulling their weight. The Pareto optimality results across different computational budgets show this isn't just good at one operating point but scales gracefully.

**Weaknesses:**

My main concern is around how much this approach depends on having really good formal proof data to begin with. The exploratory transformation is clever, but it needs high-quality theorem-proving datasets like NuminaMath-Lean to work. I'm wondering what happens when you want to explore mathematical areas that aren't well-covered in existing formal libraries does the quality drop off? The authors don't really dig into this limitation much. It would've been helpful to see some analysis of how the method performs across different mathematical domains, especially ones with sparser coverage in Mathlib. Also, while they show the method can generate complex problems, there's not much discussion about controlling *what kind* of complexity you get. If you're building educational tools or want problems at specific difficulty levels, you'd need some way to steer the generation, and that's not really addressed here.

Another thing that's bugging me is the question of whether these generated problems are actually *interesting* from a mathematical perspective, not just formally valid. Sure, they're provable and they're complex, but are they elegant? Are they the kind of problems a human mathematician would find worthwhile? The explosion check prevents contradictions, which is great, but it doesn't guarantee the problems aren't trivial or weirdly artificial. The authors mention 97% correctness on a human evaluation of informalized problems, but that's just checking if the translation is accurate not whether the underlying problem is mathematically meaningful or pedagogically useful. I also would've liked to see actual wall-clock time comparisons, not just token counts. The multi-step exploration process involves lots of sequential calls to Lean 4, and I'm curious whether the latency becomes a bottleneck in practice compared to one-shot generation methods. These aren't dealbreakers, but they're gaps that future work should probably address.

**Questions:**

Some questions that I had,

1. **Domain Coverage and Generalization**: How does DExplorer perform on mathematical domains that have sparse coverage in Mathlib? Have you tested the method on areas like algebraic topology or category theory where formal libraries are less developed? I'm curious if the exploration quality degrades noticeably or if the framework is robust to these gaps.

2. **Controlling Problem Difficulty**: Is there a way to guide the exploration toward specific difficulty levels or complexity targets? For educational applications, it would be really useful to generate, say, "calculus problems suitable for first-year undergraduates" versus "research-level analysis problems." Have you experimented with any conditioning mechanisms beyond just adjusting the step limit?

3. **Mathematical Interestingness**: Beyond formal validity, how do you assess whether generated problems are mathematically interesting or elegant? Have you considered getting feedback from professional mathematicians on whether these problems would be considered "good" problems? I'm thinking about the difference between a problem that's technically correct versus one that reveals some deeper insight or has pedagogical value.

4. **Failure Mode Analysis**: The 3.17% of statements that fail the final typecheck is interesting. What patterns do these failures follow? Are they concentrated in certain mathematical domains or with specific types of tactics? Understanding what goes wrong could really help improve the framework.

5. **Computational Efficiency in Practice**: You show impressive token cost reductions, but what about actual wall-clock time? How does the latency of making sequential Lean 4 calls compare to one-shot generation followed by proving? Is there a sweet spot in terms of step limits where you get the best quality-to-time tradeoff?

6. **Scaling the Training Data**: The exploratory transformation works great on existing proof data, but could you bootstrap the process? For example, could problems generated by DExplorer be fed back as training data after human verification, creating a self-improving cycle?

---

> ### Author Response · Authors · 2025-11-30
>
> Thank you for your appreciation of this paper and your insightful review. We prepare to update (U) the following points in the paper.
> - U1. Refined Appendix A with discussions (R6).
> - U2. Added rubrics-based reward model and LLM-as-a-judge evaluation to Appendix D.
> - U3. Add discussions (R3, R5, R9) about limitations and future works to Appendix I.
>
> Below are the detailed responses (R) to your invaluable suggestions and comments. Responses do not strictly follow the original order of Weaknesses and Questions because some new metrics have been introduced.

---

> ### Author Response · Authors · 2025-11-30
>
> > Weakness 2.1. Whether these generated problems are actually interesting from a mathematical perspective
>
> > Question 3.1. **Mathematical Interestingness**: Whether generated problems are mathematically interesting or elegant?
>
> R1. Good question! Yes. The generated problems are evaluated using carefully curated rubrics for mathematical value. In short, across evaluation settings, our method shows superior or on-par performance to all baselines and ablations. This evaluation is added to Appendix D,
>
> We conducted a rubric-based evaluation on *clarity*, *difficulty*, *elegance*, and *interestingness*.
> We employ a state-of-the-art reward model (`Skywork-Reward-V2-Llama-3.1-8B-40M`) and a generalist LLM (`DeepSeek-V3.2-Exp`) as judges to reduce bias. The LLM judges score the statements on a scale of 0-10 based on detailed rubrics provided in Appendix G.6. **Bold** highlights the best values, and *italic* indicates the second-best values.
>
> | Metric              | Method             |          |  Overall  |          |  Top-500  |          |
> | ------------------- | ------------------ | -------- | :-------: | :------: | :-------: | :------: |
> |                     |                    | Valid    |    RM     |   LLM    |    RM     |   LLM    |
> | **Clarity**         | MUSTARD            | 3791     |   13.33   |   2.95   |   21.97   |   7.47   |
> |                     | PromptCoT-DS       | 457      |   18.79   |   1.45   |   18.79   |   1.45   |
> |                     | PromptCoT-QwQ      | 1024     |   19.95   |   2.37   |   25.73   |   4.57   |
> |                     | ScaleQuest-Math    | 2035     |   14.66   |   2.24   |   23.34   |   5.39   |
> |                     | Conjecture-Prover  | 1164     | **20.67** | **3.76** |   27.11   |   7.08   |
> |                     | DExplorer (Staged) | 2340     |   19.66   |   3.31   |   29.21   |   8.02   |
> |                     | **DExplorer**      | **2726** |  *20.40*  |  *3.32*  | **30.39** | **8.14** |
> | **Difficulty**      | MUSTARD            | 3791     |   -0.80   |   0.35   |   8.86    |   1.28   |
> |                     | PromptCoT-DS       | 457      |   11.12   |   2.38   |   11.12   |   2.38   |
> |                     | PromptCoT-QwQ      | 1024     |   12.35   |   2.37   |   19.16   |   3.90   |
> |                     | ScaleQuest-Math    | 2035     |   1.01    |   0.46   |   11.30   |   1.50   |
> |                     | Conjecture-Prover  | 1164     | **13.39** |   2.62   |   21.77   |   4.90   |
> |                     | DExplorer (Staged) | 2340     |   12.25   |   2.43   |   23.50   |   6.09   |
> |                     | **DExplorer**      | **2726** |  *13.12*  | **2.63** | **24.74** | **6.74** |
> | **Elegance**        | MUSTARD            | 3791     |   7.30    |   0.52   |   18.85   |   1.68   |
> |                     | PromptCoT-DS       | 457      |   15.71   |   1.90   |   15.71   |   1.90   |
> |                     | PromptCoT-QwQ      | 1024     |   17.07   |   2.16   |   24.00   |   3.38   |
> |                     | ScaleQuest-Math    | 2035     |   7.18    |   0.59   |   18.59   |   1.68   |
> |                     | Conjecture-Prover  | 1164     | **18.68** | **2.88** |   27.18   |   5.32   |
> |                     | DExplorer (Staged) | 2340     |   17.46   |   2.65   |   29.07   |   6.58   |
> |                     | **DExplorer**      | **2726** |  *18.15*  |  *2.79*  | **30.16** | **7.09** |
> | **Interestingness** | MUSTARD            | 3791     |   -4.87   |   0.13   |   2.33    |   1.01   |
> |                     | PromptCoT-DS       | 457      |   8.40    | **2.23** |   8.40    |   2.23   |
> |                     | PromptCoT-QwQ      | 1024     | **9.01**  |  *2.07*  |   15.73   |   3.23   |
> |                     | ScaleQuest-Math    | 2035     |   -2.31   |   0.27   |   6.20    |   1.09   |
> |                     | Conjecture-Prover  | 1164     |   8.50    |   1.83   |   17.00   |   3.34   |
> |                     | DExplorer (Staged) | 2340     |   7.71    |   1.79   |   19.15   |   4.25   |
> |                     | **DExplorer**      | **2726** |  *8.76*   |   1.97   | **20.53** | **4.79** |
>
> DExplorer significantly outperforms all baselines in the top-500 regime across most metrics, particularly in *Elegance* and *Interestingness*, demonstrating its capability to discover mathematically meaningful problems.
> While our overall scores are occasionally lower than Conjecture-Prover, this is expected as DExplorer generates a vastly larger volume of valid statements ($2726$ vs. $1164$), including simpler introductory lemmas that naturally dilute the average. However, the high scores in the Top-500 set confirm that DExplorer successfully pushes the frontier of mathematical discovery.

---

> > ### Author Response · Authors · 2025-11-30
> >
> > > Question 3.2. Have you considered getting feedback from professional mathematicians on whether these problems would be considered "good" problems?
> >
> > R2. Thank you for this kind suggestion. The collaborators include graduates in mathematics, and they find many of the generated problems to be "good", "nontrivial", and "interesting". Some case studies are available in Appendix G.
> >
> > Honestly, we have wondered about this too, but do not reach out to professors in mathematics for evaluation, since the current version of DExplorer generates problems up to high school level. On the other hand, Rubrics-based RM and LLM evaluation in [R1] are more objective, reproducible and scalable. We hope they can shed some more light on the results.
> >
> > Thank you again for this suggestion :) We will continue to investigate this. Future works covering a broader domain (see R3) will invite professional mathematicians to evaluate.

---

> ### Author Response · Authors · 2025-11-30
>
> > Weakness 1.1.. What happens when you want to explore mathematical areas that aren't well-covered in existing formal libraries does the quality drop off?
>
> > Question 1. **Domain Coverage and Generalization** Whether the exploration quality degrades noticeably on mathematical domains that have sparse coverage in Mathlib?
>
> R3. Thank you for this insightful comment. Yes, the quality of generated problems degrades noticeably on out-of-distribution domains.
>
> As detailed in Appendix D.2. (line 1143), DExplorer is trained with conditional generation. The training set NuminaMath-Lean is labelled with `problem_type` and `source`, but they are coarse-grained, noisy, and imbalanced. As for `problem_type`, 45.4% are `Algebra`, 14.7% are `unknown`/`NaN`/`Other`, and only <0.01% are Trigonometry, Functional Equations or Linear Algebra; for `source`, 43.2% are `olympiads`, and 13.7% are `unknown`.
>
> For out-of-distribution generation, DExplorer is conditioned on 5 `problem_type`: `Algebraic Topology`, `Category Theory`, `Algebraic Geometry`, `Representation Theory`, and `Analytic Number Theory`; and 4 `source`: `AOPS Forum`, `Textbook`, `PhD Qualification Exam`, and `Research Paper`. In total, 400 problem-generation episodes are run (20 per combination).
>
> The evaluation results are as follows. We curated a new metric: *Alignment (LLM)*, a rubric-based LLM evaluation of the alignment between the generated problem and the generation conditions. `OOD` represents the out-of-distribution experiment, and `Main` represents the main results.
>
> OOD generation shows significant degradation across almost all metrics, especially *Alignment (LLM)*. We have manually checked the generation and found that those conditioned on `Algebraic Topology`, `Category Theory`, and `Algebraic Geometry` are more like high-school algebra or inequalities problems, and those conditioned on `Analytic Number Theory` are more like elementary number theory problems.
>
>
> | Experiment | Proven     | Contra.    | Valid      | Intract.  | Token Cost | Cplx.           | Alignment (LLM) |
> | ---------- | ---------- | ---------- | ---------- | --------- | ---------- | --------------- | --------------- |
> | OOD        | 45.50%     | 13.19%     | 86.81%     | 0.75%     | 13751      | 440.5419355     | 0.765822785     |
> | Main       | **64.18%** | **15.00%** | **84.95%** | **1.20%** | **8841**   | **515.2419355** | **3.140865737** |
>
> | Metric          | Experiment | Overall   |          |
> | --------------- | ---------- | --------- | -------- |
> |                 |            | RM        | LLM      |
> | Clarity         | OOD        | **21.06** | 2.63     |
> | Clarity         | Main       | 20.40     | **3.32** |
> | Difficulty      | OOD        | **13.14** | 2.28     |
> | Difficulty      | Main       | 13.12     | **2.63** |
> | Elegance        | OOD        | **18.45** | 2.32     |
> | Elegance        | Main       | 18.15     | **2.79** |
> | Interestingness | OOD        | 8.58      | 1.83     |
> | Interestingness | Main       | **8.76**  | **1.97** |
>
> We will add out-of-distribution generation and expanding domain coverage to Appendix I (Limitations and Future Work):
> - **Generalization to Out-of-distribution Domains.** Generalist LLMs are pretrained on web-scale knowledge, whereas the DExplorer is fine-tuned to explore mathematics primarily within the distribution of the training set. Future work may investigate techniques to maintain the broad knowledge coverage of pretrained LLMs while adapting them to formally verified deductive exploration.

---

> ### Author Response · Authors · 2025-11-30
>
> > Weakness 1.2. Controlling what kind of complexity you get.
>
> > Question 2.1.1 **Controlling Problem Difficulty**: Is there a way to guide the exploration toward specific difficulty levels or complexity targets?
>
> R4. Yes. Restricting DExplorer only to submit a conclusion after $T$ steps provides coarse-grained complexity control. If the current consumed budget is less than $T$ steps and the model generates a submitting step, it is rejected, and the model is prompted to generate a deducing step.
>
> The experiment results are as follows, where $T=20$ is used in `LimitMinStep`.  This restriction results in overall improvements at the expense of slightly fewer valid problems. Interestingly, `LimitMinStep` shows a lower average token cost for each valid generation (*Token Cost*). We hypothesize that this is because fewer tokens are wasted by contradictory generations (*Contra.*).
>
> | Experiment   | Proven   | Contra. | Valid    | Token Cost | Intract. | Cplx.           | Cplx.500     |
> | ------------ | -------- | ------- | -------- | ---------- | -------- | --------------- | ------------ |
> | LimitMinStep | **2919** | **321** | **2598** | **8242**   | 51       | **544.7000393** | **1373.646** |
> | Main         | 3209     | 497     | 2726     | 8841       | **60**   | 515.2419355     | 1373.554     |
>
> | Metric          | Experiment   | Overall   |          | Top-500   |            |
> | --------------- | ------------ | --------- | -------- | --------- | ---------- |
> |                 |              | RM        | LLM      | RM-Top500 | LLM-Top500 |
> | Clarity         | LimitMinStep | **21.76** | **3.52** | **31.35** | **8.23**   |
> | Clarity         | Main         | 20.40     | 3.32     | 30.39     | 8.14       |
> | Difficulty      | LimitMinStep | **14.21** | **2.78** | **25.31** | **6.82**   |
> | Difficulty      | Main         | 13.12     | 2.63     | 24.74     | 6.74       |
> | Elegance        | LimitMinStep | **19.45** | **2.98** | **30.74** | **7.17**   |
> | Elegance        | Main         | 18.15     | 2.79     | 30.16     | 7.09       |
> | Interestingness | LimitMinStep | **10.22**     | **2.07**     | **21.93**     | **4.88**       |
> | Interestingness | Main         | 8.76      | 1.97     | 20.53     | 4.79       |
>
> > Question 2.1.2 For educational applications, it would be really useful to generate, say, "calculus problems suitable for first-year undergraduates" versus "research-level analysis problems."
>
> > Question 2.2. Have you experimented with any conditioning mechanisms beyond just adjusting the step limit?
>
> R5. As explained in R3, DExploration can condition on `problem_type` and `source`, providing a proxy for subject and difficulty. However, due to the noisy, imbalanced training data, this capability is to some extent limited. For alignment to conditions, in-domain generation results in a 3.14 LLM score, and OOD generation results in a 0.77 rubrics-based LLM score.
>
> We will add conditional generation to Appendix I (Limitations and Future Work):
> - **Controllable and Meaningful Generation**. Current DExploration is bootstrapped by supervised fine-tuning (SFT) on data distilled from existing theorem-proof pairs. While our experiments show that it generates problems with superior elegance and difficulty compared to baselines, it relies on implicit imitation rather than explicit optimization. Consequently, it underperforms at controlling specific domains, knowledge points, or difficulty levels on demand. Future works may incorporate conditional training variables or reinforcement learning (RL) with rubric-based reward functions to achieve precise control and further maximize the meaningfulness of the generation.

---

> > ### Author Response · Authors · 2025-11-30
> >
> > > Question 4. **Failure Mode Analysis**: The 3.17% of statements that fail the final typecheck is interesting. What patterns do these failures follow? Are they concentrated in certain mathematical domains or with specific types of tactics? Understanding what goes wrong could really help improve the framework.
> >
> >
> > R6. Thank you for this insightful comment. We conducted a thorough investigation to resolve this. After careful debugging, the remaining 3.17% of samples that were submitted but failed in statement-proof reconstruction are all resolved.
> >
> > The error types and our fixes are as follows:
> >
> > - 44.76%: Failure to transform pretty-printed `Expr` back to `Syntax`. For example, `example : (∑ (i : Fin 10), (i : ℚ)) / 2 = 22.5 := sorry` results in the proof state
> >     `⊢ (∑ i : Fin 10, ↑↑i) / 2 = 22.5`
> >  Directly reconstructed statement `example : (∑ i : Fin 10, ↑↑i) / 2 = 22.5 := sorry` fails with "failed to synthesize AddCommMonoid Float". A correct reconstruction can be `example : (∑ i : Fin 10, (↑↑i : ℚ)) / 2 = 22.5 := sorry` (adding type ascription to `i`).
> >  We resolve this by optimizing pretty-printer settings and delaborators.
> > - 40.00%: Deducing steps introduce new variables.
> >  Tactics like  `obtain`,  `cases'`, `rcases` sometimes introduce new variables that are not tracked in the introduction history. If these are used in the submitting step, the recomposed statement might be incorrect.
> > For example, an erroneous problem generation process is:
> > ```lean4
> > have φ : ℝ → ℝ := sorry -- Introduce
> >
> > open Real Set in
> > have hφ : φ = fun t => exp t * (cos t + sin t) := sorry  -- Introduce
> >
> > have ψ : ℝ → ℝ := sorry -- Introduce
> >
> > open Real in
> > have hψ : ψ = fun t => exp t * (cos t - sin t) := sorry -- Introduce
> >
> > have hψ' : ∀ t, ∃ s, ψ t = s * φ t := sorry -- Introduce
> >
> > have h0 := hψ' 0 -- Deduce
> >
> > simp [hφ, hψ] at h0 -- Deduce
> >
> > have h1 := hψ' (Real.pi / 2) -- Deduce, h1 : ∃ s, -rexp (π / 2) = s * rexp (π / 2)
> >
> > simp [hψ, hφ] at h1 -- Deduce
> >
> > rcases h1 with ⟨s, hs⟩ -- Deduce (Introducing new variables)
> >
> > open Real in
> > have h1 : s = -1 := by -- Deduce
> >  apply (mul_left_inj' (exp_ne_zero (π / 2))).mp
> >  linarith
> >
> > submit_answer h1 -- Submit
> > ```
> > The reconstructed statement is:
> > ```lean4
> > open Set Real
> > example
> > (φ : ℝ → ℝ)
> > (hφ : φ = fun t => exp t * (cos t + sin t))
> > (ψ : ℝ → ℝ)
> > (hψ : ψ = fun t => exp t * (cos t - sin t))
> > (hψ' : ∀ t, ∃ s, ψ t = s * φ t)
> > : s = -1 := by sorry
> > ```
> >  The reconstructed proof is:
> > ```lean4
> > have h0 := hψ' 0
> >
> > simp [hφ, hψ] at h0
> >
> > have h1 := hψ' (Real.pi / 2)
> >
> > simp [hψ, hφ] at h1
> >
> > rcases h1 with ⟨s, hs⟩ -- This step brings another `s`
> >
> > open Real in
> > have h1 : s = -1 := by
> >  apply (mul_left_inj' (exp_ne_zero (π / 2))).mp
> >  linarith
> >
> > exact h1
> > ```
> > The deducing step `rcases h1 with ⟨s, hs⟩` adds a new variable `s : ℝ` and a hypothesis `hs : -rexp (π / 2) = s * rexp (π / 2)` to the context. However, `s : ℝ` does not exist in any introducing step. Therefore, the reconstructed statement implicitly introduces `s : ℤ` as an implicit parameter.
> >
> > In the reconstructed proof, `rcases h1 with ⟨s, hs⟩` brings another `s : ℝ`. Thus, the previous implicitly introduced `s : ℤ` is anonymized into `s✝ : ℤ`, and the proving target changes to `⊢ s✝ = -1`. The subsequent tactic `have h1 : s = -1` deduces `s = -1`, but it is not equivalent to `s✝ = -1`.
> >
> > Actually, this behavior does not meet our definition of *deductive tactic* (line 229), since it modifies variables in the `Type` universe. Our initial code implementation contained a bug and failed to reject this tactic. Now we have updated the code implementation.
> >
> > - 7.62%: Introducing step anonymizes existing variables.
> > For example, an erroneous problem generation process is:
> > ```lean4
> > have a : ℝ := sorry
> >
> > have ha : a = 1 := sorry
> >
> > -- Some deducing steps
> >
> > have ha : a = 2 := sorry
> >
> > -- Remaining steps
> > ```
> > The reconstructed statement is
> > ```
> > example
> > (a : ℝ)
> > (ha : a = 1)
> > (ha : a = 2)
> > ...
> > := sorry
> > ```
> > In the problem generation process `Some deducing steps` may use `ha` with type `a = 1`. However, the initial proof state of the reconstructed statement is
> > ```lean4
> > a : ℝ
> > ha✝ : a = 1
> > ha : a = 2
> > ...
> > ```
> > Therefore, in the reconstructed proof, the `ha` used by `Some deducing steps` is `a = 2`, the original `ha : a = 1` is anonymized into `ha✝ : a = 1`. Hence, the reconstructed proof fails.
> >
> > We resolve this issue by strictly prohibiting steps that anonymize existing variables/hypotheses.

---

> ### Author Response · Authors · 2025-11-30
>
> - 7.62%: Tactics behave differently between problem generation and proof reconstruction. This addresses your concern that "Lean tactics are tricky." Our solution is to encapsulate tactics into the same context as problem generation. Concretely, Given a deductive step $s$ that transforms proof state $\Gamma \vdash \texttt{False}$ to $\Gamma^\prime \vdash \texttt{False}$ during exploration, the operator $\text{Encapsulate}(s, \Gamma^\circ, \Gamma, \Gamma^\prime)$ represents the following Lean tactic sequence. It temporarily restricts the proof state to match $\Gamma$, executes $s$, and then restores the wider context.
> Denote the sets of declarations involved as:
> - *Future Variables*: $\Gamma^\circ \setminus \Gamma = \{(v_i : T_i)\}_{i=1}^n$ (variables introduced after the step $s$).
> - *New Deductions*: $\Gamma^\prime \setminus \Gamma = \{(h^\prime_i : H^\prime_i)\}_{i=1}^p$.
> - *Consumed Hypotheses*: $\Gamma \setminus \Gamma^\prime = \{(h_i : H_i)\}_{i=1}^q$ (hypotheses consumed/cleared by $s$).
> ```
> have _H_DEDUCTION_ENCAPSULATION := by
>   -- The proof target U is hidden
>   clear v_1 v_2 ... v_n
>   -- Future Variables Γ° \ Γ are removed
>   -- The context is now identical to Γ
>   s
>   -- The context is now identical to Γ'
>   exact (show (H'_1 ∧ H'_2 ∧ ... ∧ H'_p) by exact ⟨h'_1, h'_2, ..., h'_p⟩)
> -- The derived facts are consolidated to `_H_DEDUCTION_ENCAPSULATION'
> clear h'_1 h'_2 ... h'_p h_1 h_2 ... h_q
> rcases _H_DEDUCTION_ENCAPSULATION with h'_1, h'_2, ..., h'_p
> -- Apply changes from Γ to Γ' in the current proof state
> ```
> Therefore, during proof reconstruction, $\text{Encapsulate}(s, \Gamma^\circ, \Gamma, \Gamma^\prime)$ ensures $s$ to behave identically to problem generation.
>
> More discussions, explanations, and proofs can be found in Appendix A in the revision.
>
>
> > Weakness 2.2. Actual wall-clock time comparisons.
>
> > Question 5.1. **Computational Efficiency in Practice**: What about actual wall-clock time? How does the latency of making sequential Lean 4 calls compare to one-shot generation followed by proving?
>
> R7. It is difficult to provide a precise comparison of each method's wall-clock time because our computational cluster was unstable and other users frequently caused IO/CPU congestion. We analyze the raw experiment logs and provide a rough comparison of total experiment time as follows.
>
> - MUSTARD: -
> 	- Informal Problem Generation: Not reported in the original paper
> 	- Statement & Proof Autoformalization:  Not reported in original paper
> - PromptCoT-QwQ: > 238,333 seconds
> 	- Informal Problem Generation: Not reported in the original paper
> 	- Autoformalization: GPU=4, CPU=96 | 858 seconds
> 	- Proving: GPU=3, CPU=60 | 237,475 seconds
> - PromptCoT-DS: >289,012 seconds
> 	- Informal Problem Generation: Not reported in the original paper
> 	- Autoformalization: GPU=4, CPU=96 | 909 seconds
> 	- Proving: GPU=3, CPU=60 | 288,103 seconds
> - ScaleQuest-Math: > 23,050 seconds
> 	- Informal Problem Generation: Not reported in the original paper
> 	- Autoformalization: GPU=4, CPU=96 | 992 seconds
> 	- Proving: GPU=3, CPU=60 | 22,058 seconds
> - Conjecture-Prover: 110,251 seconds
> 	- Statement Generation + Proving: 2 GPUs for statement generation, 6 GPUs for proving, CPU=96 | 110,251 seconds
> - DExplorer (Staged): 35,905 seconds
> 	- DExploration + Statement/Proof Reconstruction: 4 GPUs for DExplorer, 4 GPUs for DeepSeek-Prover-V2-7B (nonCoT), CPU=96 | 35,905 seconds
> - DExplorer: 60,143 seconds
> 	- DExploration + Statement/Proof Reconstruction: 4 GPUs for DExplorer, 4 GPUs for DeepSeek-Prover-V2-7B (nonCoT), CPU=96 | 60,143 seconds
>
> The average time consumption of DExplorer is significantly lower than that of Conjecture-Prover, PromptCoT-QwQ, and PromptCoT-DS. As for ScaleQuest-Math, although its wall clock time is shorter, its generated statements are significantly easier.
>
> The discrepancy between DExplorer and DExplorer (Staged) is likely due to varying loads on the shared cluster during different runs.

---

> > ### Author Response · Authors · 2025-11-30
> >
> > > Question 5.2. Is there a sweet spot in terms of step limits where you get the best quality-to-time tradeoff?
> >
> > R8. Trading-off quality and time is fairly complicated, since "quality" involves multiple objectives. Please refer to Sec. 5.3.1 (line 421) "Pareto-Optimality" and Appendix C, where we have experimented adjusting the step limit $N_s$ among $\{4, 8, 16, 32, 80\}$ for DExplorer.
> >
> > Token cost is a proxy metric for time consumption. Our preliminary benchmarking shows that proof search and whole-proof generation are mainly bound by GPU instead of CPU if highly parallelized (assuming GPU:CPU = 1:24).
> >
> > As Fig. 4 demonstrates, as the step limit increases, the portion of valid generations among all attempts (*success rate*) increases in a log-linear manner. However, the distribution of *proof complexity* and *item difficulty* significantly shifts from easy to hard as $N_s$ increases. The average complexities of $N_s=[4, 8, 16, 32, 80]$ are $[230, 311, 483, 513, 515]$. The average token costs are $[6300, 4480, 4565, 5925, 8841]$.
> >
> > If focused on the average token cost per valid generation, setting $N_s=8$ is optimal, with an average token cost of $4480$. If also considering the proof complexity, $N_s=16, 32$ are better. When taking difficulty, *Intract.* (the number of unproven statements for SOTA LLM provers) and rubric-based metrics, the situation becomes more complicated, and the optimal step limit differs by the optimization objective.

---

> > > ### Author Response · Authors · 2025-11-30
> > >
> > > > Question 6. **Scaling the Training Data**: The exploratory transformation works great on existing proof data, but could you bootstrap the process? For example, could problems generated by DExplorer be fed back as training data after human verification, creating a self-improving cycle?
> > >
> > > R9. Excellent idea! Actually, we have thought about this too. However, preparing and cleaning datasets, designing scalable filtering strategies and/or reward functions, and implementing an expert iteration or RL codebase require significant time, effort, and resources. Given the limited time during the discussion phase and our computational resource limitations, we do not yet have conclusive data to verify this idea.
> > >
> > > We will add the following points to Appendix I (Limitations and Future Work) and continue to investigate this.
> > > 1. Controllable and Meaningful Generation. Current DExploration is bootstrapped by supervised fine-tuning (SFT) on data distilled from existing theorem-proof pairs. While our experiments show that it generates problems with superior elegance and difficulty compared to baselines, it relies on implicit imitation rather than explicit optimization. Consequently, it underperforms at controlling specific domains, knowledge points, or difficulty levels on demand. Future works may incorporate conditional training variables or **reinforcement learning (RL) with rubric-based reward functions** to achieve precise control and further maximize the meaningfulness of the generation.
> > > 2. **Co-evolution of Formal and Informal Reasoning.** DExploration effectively generates complex and difficult problems. This creates an opportunity for a virtuous cycle: using synthesized problems to adversarially train and improve formal provers or informal reasoners. Future work may explore this co-evolutionary pipeline to simultaneously push the boundaries of problem generation and automated reasoning.
> > >
> > > As for the second *Adversarial training*, our response to Reviewer 8iAd [R1] provides some preliminary results on using statements generated by DExplorer to improve theorem proving performance.
> > >
> > > LLM provers trained with GRPO on the 2726 generated problems without specifically tuned hyperparameters show consistently improved performance. The performance of `DSP-V1.5` is significantly lower than the reported performance in their paper (and in our previous experiments). This might be caused by chat template mismatch of our RL framework verl.
> > >
> > > | Bench        | Exp                         | Pass@1     | Pass@4     | Pass@8     | Pass@32    |
> > > | ------------ | --------------------------- | ---------- | ---------- | ---------- | ---------- |
> > > | MiniF2F-test | DSP-V1.5-7B                 | 4.92%      | 10.25%     | 15.98%     | 31.97%     |
> > > |              | **DSP-V1.5-7B + RL**        | **29.92%** | **39.34%** | **42.62%** | **46.31%** |
> > > |              | DSP-V2-7B (nonCoT)          | 50.82%     | 61.89%     | 62.30%     | 64.34%     |
> > > |              | **DSP-V2-7B (nonCoT) + RL** | **58.61%** | **63.52%** | **64.75%** | **66.39%** |
> > > | ProofNet     | DSP-V1.5-7B                 | 3.50%      | 7.55%      | 9.43%      | 12.94%     |
> > > |              | **DSP-V1.5-7B + RL**        | **8.36%**  | **10.24%** | **11.59%** | **14.29%** |
> > > |              | DSP-V2-7B (nonCoT)          | 15.36%     | 20.49%     | 21.29%     | 22.37%     |
> > > |              | **DSP-V2-7B (nonCoT) + RL** | **19.95%** | **21.56%** | **22.64%** | **22.91%** |
> > > | PutnamBench  | DSP-V1.5-7B                 | 0          | 1          | 1          | 1          |
> > > |              | DSP-V1.5-7B + RL            | **2**      | **4**      | **4**      | **6**      |
> > > |              | DSP-V2-7B (nonCoT)          | **6**      | **6**      | 6          | **9**      |
> > > |              | DSP-V2-7B (nonCoT) + RL     | 5          | **6**      | **7**      | 8          |
> > >
> > > We hope these will shed some more light on this excellent idea. Thank you again for this suggestion :)

---

### Official Review · Reviewer_kBGE · 2025-10-30

**Soundness:** 2
**Presentation:** 3
**Contribution:** 2
**Rating:** 2
**Confidence:** 3

**Summary:**

The authors introduce DExploration, a step-by-step problem generation framework grounded in Lean 4, designed to scale synthetic mathematical data generation while ensuring provability. This approach offers a principled solution to the trade-off between scalability and formal correctness in mathematical reasoning datasets. Their empirical results show notable improvements in proof success rates, problem difficulty, and efficiency (token cost), though the study’s evaluation setup and comparison baselines raise questions about the robustness and fairness of the reported gains.

**Strengths:**

- Casting problem synthesis as an exploration process with three atomic actions: Introduce, Deduce, and Submit, is very simple and realistic, which closely mirrors the workflow of mathematicians, as well as a good way to approach the trilemma mentioned.
- Reconstructing a formally checkable Lean 4 statement and proof from the agent’s exploration trajectory ensures validity in a principled, rather than heuristic way.
- The Exploratory Transformation procedure (deductive rewriting → dependency graph → topological ordering) provides a sensible means to convert existing proofs into trajectories for supervised fine-tuning. The depth-prioritised reassembly is intuitively well-founded.
- The paper reports substantial improvements in proof success and validity rates, alongside major reductions in token cost, which could be considered the main contribution of the paper. The generation of novel statements unprovable by SOTA provers further highlights the method’s potential (though if the results are robust).

**Weaknesses:**

- Lean tactics are tricky and very dependent on context. Common ones like linarith, nlinarith, apply, constructor, and cases can behave unpredictably in different settings. The authors note themselves (around line 1007) that a formal treatment of “deductive tactics” is still future work, and that some tactics can occasionally break the intended constraints. This makes the “guaranteed provability” claim a bit weaker than stated. Without a machine-checkable proof that the tactic set always preserves the invariants in their lemmas, the claim remains conditional. The reported 96.83% success rate is impressive, but it also shows that the guarantee isn’t absolute yet.
- The method relies on heuristic or one-off checks, which are inherently probabilistic; some contradictions could slip through, and some valid but complex cases might get rejected. The paper mentions this, but doesn’t go into detail about how often such cases occur or how sensitive the results are to the choice of prover.
- The manual evaluation covers 100 samples, with 97 marked correct. That’s a good sign, but the sample is quite small compared to the 2,726 valid generations and 39k training proofs. Also, the evaluation only checks for correctness, not for clarity or difficulty of the generated proofs.
- The statement that “DExploration eliminates reliance on external provers and achieves fully provable generation” feels too strong. The system still depends on Lean, Aesop, and LLM-based provers for verification, and about 3% of outputs fail the final check. It would be fairer to say that it largely reduces external dependencies rather than eliminating them.
- Finally, the paper doesn’t include a limitations section. It would help the overall credibility to acknowledge open issues; such as tactic stability, evaluation scope, or remaining failure modes.

**Questions:**

On tactic reliability:
- How do the authors ensure that all Lean tactics used (e.g., linarith, nlinarith, apply, constructor, cases) preserve the invariants required by their proofs?
- Could they provide a more formal or machine-checkable justification that the selected tactic set does not violate the provability constraint?
- Are there examples where tactic behavior led to incorrect or unverifiable results?

On heuristic verification:
- The approach seems to rely on heuristic or single-sample checks. Have the authors quantified how often these checks fail to catch contradictions or reject valid cases?
- How sensitive are the reported results to the choice of prover or verification strategy?

On evaluation design:
- Why was a sample of 100 items chosen for manual evaluation, given the scale of generated proofs?
- Could the authors provide metrics beyond binary correctness—such as clarity, ambiguity, or problem difficulty?
- How representative is the 100-sample subset of the full set of 2,726 valid generations?

On the claim of “eliminating” external provers:
- Since the system still depends on Lean, Aesop, and LLM-based provers for validation, could the authors clarify what they mean by “eliminates reliance”?
- Would it be more accurate to describe the approach as “reducing” rather than “eliminating” dependency on external provers?

On limitations and open issues:
- Could the authors discuss current limitations, such as tactic reliability, scalability, or remaining failure modes?
- What are the main barriers to achieving a fully formalised notion of “deductive tactics”?

---

> ### Author Response · Authors · 2025-11-30
>
> We greatly appreciate your constructive comments, as well as your time and effort in this invaluable review! Briefly, we have updated (U) the following points in the revision:
> - U1. Refined Appendix A with formal proofs and discussions (referring to Responses R1-R3).
> - U2. Added rubrics-based reward model and LLM-as-a-judge evaluation to Appendix D.
> - U3. Add discussions about limitations and future works to Appendix I.
>
> Since the weaknesses and questions correlate with each other, our response (R) is organized by topic. If you find any missing or unclear points, welcome for further discussion :)
>
> > Weakness 1.1. (around line 1007) that a formal treatment of “deductive tactics” is still future work
>
> R1. We sincerely apologize for the confusion caused by the leftover comment. Line 1007 in Appendix A was a finished TODO that was accidentally uncommented in the submission. The definition of deductive tactics is provided in lines 228-230.
>
> > Weakness 1.2. some tactics can occasionally break the intended constraints.
>
> > Question 1.1. How do the authors ensure that all Lean tactics used (e.g., linarith, nlinarith, apply, constructor, cases) preserve the invariants required by their proofs
>
> > Question 1.3. Are there examples where tactic behavior led to incorrect or unverifiable results?
>
> R2. Thank you for this comment! It sparked a productive discussion and re-examination among the authors.
> TL;DR: After careful debugging, the remaining 3.17% of samples that were submitted but failed in statement-proof reconstruction are all resolved.
> The error types and our fixes are as follows:
>
> - 44.76%: Failure to transform pretty-printed `Expr` back to `Syntax`. For example, `example : (∑ (i : Fin 10), (i : ℚ)) / 2 = 22.5 := sorry` results in the proof state
>     `⊢ (∑ i : Fin 10, ↑↑i) / 2 = 22.5`
>  Directly reconstructed statement `example : (∑ i : Fin 10, ↑↑i) / 2 = 22.5 := sorry` fails with "failed to synthesize AddCommMonoid Float". A correct reconstruction can be `example : (∑ i : Fin 10, (↑↑i : ℚ)) / 2 = 22.5 := sorry` (adding type ascription to `i`).
>  We resolve this by optimizing pretty-printer settings and delaborators.
> - 40.00%: Deducing steps introduce new variables.
>  Tactics like  `obtain`,  `cases'`, `rcases` sometimes introduce new variables that are not tracked in the introduction history. If these are used in the submitting step, the recomposed statement might be incorrect.
> For example, an erroneous problem generation process is:
> ```lean4
> have φ : ℝ → ℝ := sorry -- Introduce
>
> open Real Set in
> have hφ : φ = fun t => exp t * (cos t + sin t) := sorry  -- Introduce
>
> have ψ : ℝ → ℝ := sorry -- Introduce
>
> open Real in
> have hψ : ψ = fun t => exp t * (cos t - sin t) := sorry -- Introduce
>
> have hψ' : ∀ t, ∃ s, ψ t = s * φ t := sorry -- Introduce
>
> have h0 := hψ' 0 -- Deduce
>
> simp [hφ, hψ] at h0 -- Deduce
>
> have h1 := hψ' (Real.pi / 2) -- Deduce, h1 : ∃ s, -rexp (π / 2) = s * rexp (π / 2)
>
> simp [hψ, hφ] at h1 -- Deduce
>
> rcases h1 with ⟨s, hs⟩ -- Deduce (Introducing new variables)
>
> open Real in
> have h1 : s = -1 := by -- Deduce
>  apply (mul_left_inj' (exp_ne_zero (π / 2))).mp
>  linarith
>
> submit_answer h1 -- Submit
> ```
> The reconstructed statement is:
> ```lean4
> open Set Real
> example
> (φ : ℝ → ℝ)
> (hφ : φ = fun t => exp t * (cos t + sin t))
> (ψ : ℝ → ℝ)
> (hψ : ψ = fun t => exp t * (cos t - sin t))
> (hψ' : ∀ t, ∃ s, ψ t = s * φ t)
> : s = -1 := by sorry
> ```
>  The reconstructed proof is:
> ```lean4
> have h0 := hψ' 0
>
> simp [hφ, hψ] at h0
>
> have h1 := hψ' (Real.pi / 2)
>
> simp [hψ, hφ] at h1
>
> rcases h1 with ⟨s, hs⟩ -- This step brings another `s`
>
> open Real in
> have h1 : s = -1 := by
>  apply (mul_left_inj' (exp_ne_zero (π / 2))).mp
>  linarith
>
> exact h1
> ```
> The deducing step `rcases h1 with ⟨s, hs⟩` adds a new variable `s : ℝ` and a hypothesis `hs : -rexp (π / 2) = s * rexp (π / 2)` to the context. However, `s : ℝ` does not exist in any introducing step. Therefore, the reconstructed statement implicitly introduces `s : ℤ` as an implicit parameter.
>
> In the reconstructed proof, `rcases h1 with ⟨s, hs⟩` brings another `s : ℝ`. Thus, the previous implicitly introduced `s : ℤ` is anonymized into `s✝ : ℤ`, and the proving target changes to `⊢ s✝ = -1`. The subsequent tactic `have h1 : s = -1` deduces `s = -1`, but it is not equivalent to `s✝ = -1`.
>
> Actually, this behavior does not meet our definition of *deductive tactic* (line 229), since it modifies variables in the `Type` universe. Our initial code implementation contained a bug and failed to reject this tactic. Now we have updated the code implementation.

---

> > ### Author Response · Authors · 2025-11-30
> >
> > - 7.62%: Introducing step anonymizes existing variables.
> > For example, an erroneous problem generation process is:
> > ```lean4
> > have a : ℝ := sorry
> >
> > have ha : a = 1 := sorry
> >
> > -- Some deducing steps
> >
> > have ha : a = 2 := sorry
> >
> > -- Remaining steps
> > ```
> > The reconstructed statement is
> > ```
> > example
> > (a : ℝ)
> > (ha : a = 1)
> > (ha : a = 2)
> > ...
> > := sorry
> > ```
> > In the problem generation process `Some deducing steps` may use `ha` with type `a = 1`. However, the initial proof state of the reconstructed statement is
> > ```lean4
> > a : ℝ
> > ha✝ : a = 1
> > ha : a = 2
> > ...
> > ```
> > Therefore, in the reconstructed proof, the `ha` used by `Some deducing steps` is `a = 2`, the original `ha : a = 1` is anonymized into `ha✝ : a = 1`. Hence, the reconstructed proof fails.
> >
> > We resolve this issue by strictly prohibiting steps that anonymize existing variables/hypotheses.
> >
> > - 7.62%: Tactics behave differently between problem generation and proof reconstruction. This addresses your concern that "Lean tactics are tricky." Our solution is to encapsulate tactics into the same context as problem generation. Concretely, Given a deductive step $s$ that transforms proof state $\Gamma \vdash \texttt{False}$ to $\Gamma^\prime \vdash \texttt{False}$ during exploration, the operator $\text{Encapsulate}(s, \Gamma^\circ, \Gamma, \Gamma^\prime)$ represents the following Lean tactic sequence. It temporarily restricts the proof state to match $\Gamma$, executes $s$, and then restores the wider context.
> > Denote the sets of declarations involved as:
> > - *Future Variables*: $\Gamma^\circ \setminus \Gamma = \{(v_i : T_i)\}_{i=1}^n$ (variables introduced after the step $s$).
> > - *New Deductions*: $\Gamma^\prime \setminus \Gamma = \{(h^\prime_i : H^\prime_i)\}_{i=1}^p$.
> > - *Consumed Hypotheses*: $\Gamma \setminus \Gamma^\prime = \{(h_i : H_i)\}_{i=1}^q$ (hypotheses consumed/cleared by $s$).
> > ```
> > have _H_DEDUCTION_ENCAPSULATION := by
> >   -- The proof target U is hidden
> >   clear v_1 v_2 ... v_n
> >   -- Future Variables Γ° \ Γ are removed
> >   -- The context is now identical to Γ
> >   s
> >   -- The context is now identical to Γ'
> >   exact (show (H'_1 ∧ H'_2 ∧ ... ∧ H'_p) by exact ⟨h'_1, h'_2, ..., h'_p⟩)
> > -- The derived facts are consolidated to `_H_DEDUCTION_ENCAPSULATION'
> > clear h'_1 h'_2 ... h'_p h_1 h_2 ... h_q
> > rcases _H_DEDUCTION_ENCAPSULATION with h'_1, h'_2, ..., h'_p
> > -- Apply changes from Γ to Γ' in the current proof state
> > ```
> > Therefore, during proof reconstruction, $\text{Encapsulate}(s, \Gamma^\circ, \Gamma, \Gamma^\prime)$ ensures $s$ to behave identically to problem generation.
> >
> > More discussions, explanations, and proofs can be found in Appendix A in the revision.
> >
> >
> > > Question 1.2. Could they provide a more formal or machine-checkable justification that the selected tactic set does not violate the provability constraint?
> >
> > R3. Sure. Given the above encapsulate, by induction, we can prove that for the $k$-th deduction step, the proof state after execuuting it during proof reconstruction is $\Gamma^{(k+1)} \vdash U$, and $\Gamma^{(k+1)}$ is a superset of the context during problem generation $\Gamma_{D_{k}+1}$. Therefore, the last tactic $\texttt{exact } h_U$ finishes the proof. The detailed proof can be found in Appendix A in the revision.

---

> ### Author Response · Authors · 2025-11-30
>
> > Weakness 2. Method relies on **heuristic or one-off checks**, which are inherently probabilistic; some contradictions could slip through, and some valid but complex cases might get rejected.
>
> > Question 2.1 How often these checks fail to catch contradictions or reject valid cases.
>
> > Question 2.2 Sensitivity to the choice of prover or verification strategy.
>
> R4. Thank you for this insightful comment! We analyze the sensitivity to the choice of prover or verification strategy from two aspects: the light-weighted explosion check, and the final evaluation.
>
>
> The lightweight explosion check in DExploration only rejects introducing steps that are proven contradictory. Therefore, it does not reject valid cases. The frequency it fails to catch contradictions is indicated by *Contra.* in Table 1, which is significantly lower than baselines without an explosion check.
>
> To measure the sensitivity of experiment results to the explosion check, we run two experiments with no explosion check (`None`) and only Aesop for the explosion check (`Aesop`). The results are as follows.
>
> | Prover         | Proven   | Contra. | Valid    | Intract. | Token Cost      | Complexity      | Cplx.500     |
> | -------------- | -------- | ------- | -------- | -------- | --------------- | --------------- | ------------ |
> | None           | **3441** | 867     | 2574     | 49       | 8085.494561     | 512.6308911     | 1293.928     |
> | Aesop          | 3417     | 809     | 2608     | 47       | **7681.771089** | **553.3174541** | **1423.288** |
> | Aesop+DeepSeek | 3209     | **497** | **2726** | **60**   | 8841.159102     | 515.2419355     | 1373.554     |
>
> As the explosion check weakens, DExplorer generates more *proven* statements and more *contradictory* statements. The number of statements that are unprovable by the three SOTA provers (*Intract.*) lowers. Using only Aesop for explosion check demonstrates the best performance on *token cost*, complexity (*Cplx.*) and top-500 complexity (*Cplx.500*). But they do not significantly differ. Notably, the portion of contradictory statements among all proven statements (*Contra.*/*Proven*) of `None` is still significantly lower than all baselines except MUSTARD, whose generated problems are significantly easier.
>
> The final evaluation, i.e., the reported *Contra.* in Table 1, does not rely on heuristic or one-off checks. As mentioned in lines 355-357, during evaluation, the contradiction check is conducted with 3 SOTA LLM provers, Goedel-Prover-V2-8B, DeepSeek-Prover-V2-7B (CoT), and Kimina-Prover-Distill-8B, with a total of $K=3\times 4$ attempts. Given a proven statement $\Gamma \vdash U$,
> 1. We first try to prove $[] \vdash \exists \Gamma$., i.e., the *satisfiability* of the statement. If this holds, the statement is not contradictory.
> 2. If the satisfiability check is not proven, we try to prove $\Gamma \circ [h_U : U] \vdash \texttt{False}$, i.e., the *contradiction*. If this holds, the statement is contradictory.
> 3. For statements whose *satisfiability* and *contradiction* are neither proven, we view them as valid.
> Therefore, the above check does not reject valid cases.
>
> To measure the sensitivity of *Contra.* evaluation, we control the number of proving attempts $K$ in the satisfiability check and contradiction. The results are as follows, which are based on the above `None` experiment. *Undecided* denotes the percentage whose *satisfiability* and *contradiction* are neither proven. *12+Human* represents manually checking the remaining 281 statements.
>
> | K             | 1      | 2      | 4      | 8      | 12     | 12+Human |
> | ------------- | ------ | ------ | ------ | ------ | ------ | -------- |
> | Satisfiable   | 36.73% | 50.10% | 60.13% | 65.07% | 66.64% | 73.58%   |
> | Contradictory | 19.35% | 22.78% | 24.24% | 25.02% | 25.20% | 26.42%   |
> | Undecided     | 43.91% | 27.11% | 15.63% | 9.91%  | 8.17%  | 0.00%    |
>
> The results validate our choice of $K=12$ is enough. The *Contradictory* plateaus at $K=8$, it only improves $0.18\%$ from $K=8$ to $K=12$. However, human evaluation of 281 *undecided* statements finds 42 are contradictory, i.e., the false positive rate of the contradictory check is 1.22%. On the other hand, if we only view those *Satisfiable* as valid, the false positive rate is $0$, at the expense of 6.95% false negatives.

---

> ### Author Response · Authors · 2025-11-30
>
> > Weakness 3.1. 100 samples covered by **manual evaluation** is too small compared to the 2,726 valid generations.
>
> > Question 3.1. **Why was a sample of 100 items chosen for manual evaluation**, given the scale of generated proofs?
>
>
> R5. To clarify, the manual evaluation is to check the correctness of *informalization*, i.e., translating formal statements into natural language. This is a significantly easier[1] task than autoformalization (translating natural language math into formal), especially given that our focused domain is up to high-school competition level. We have checked 400 more formal-informal pairs, which result in 97.25% accuracy.
>
> > Question 3.3. How **representative** is the 100-sample subset of the full set of 2,726 valid generations?
>
> R6. The 100 samples (and the subsequent 400 samples) are uniformly sampled from the 2726 generations. Therefore, they are representative enough.

---

> ### Author Response · Authors · 2025-11-30
>
> > Weakness 3.2. Evaluation only checks for correctness, not for **clarity or difficulty** of the generated proofs.
>
> > Question 3.2. Could the authors provide **metrics beyond binary correctness**—such as clarity, ambiguity, or problem difficulty?
>
> R7. Thank you for this suggestion. To clarify, difficulty is measured using *item difficulty*[2] (line 387), which is broadly used in educational assessment. It measures the proportion of respondents who successfully complete or correctly answer the item.
>
> We appreciate this constructive suggestion to provide more metrics beyond binary correctness. Generated problems are evaluated based on carefully curated rubrics on the mathematical value. In short, across evaluation settings, our method shows superior or on-par performance to all baselines and ablations. This evaluation is added to Appendix D,
>
> We conducted a rubric-based evaluation on *clarity*, *difficulty*, *elegance*, and *interestingness*.
> We employ a state-of-the-art reward model (RM), `Skywork-Reward-V2-Llama-3.1-8B-40M`, and a generalist LLM, `DeepSeek-V3.2-Exp`, as judges to reduce bias. The LLM judges score the statements on a scale of 0-10 based on detailed rubrics provided in Appendix G.6. **Bold** highlights the best values, and *italic* indicates the second-best values.
>
> | Metric              | Method             |          | Overall  |          | Top-500  |          |
> | ------------------- | ------------------ | -------- | :-------: | :------: | :-------: | :------: |
> |                     |                    | Valid    | RM     | LLM    | RM     | LLM    |
> | **Clarity**         | MUSTARD            | 3791     | 13.33   | 2.95   | 21.97   | 7.47   |
> |                     | PromptCoT-DS       | 457      | 18.79   | 1.45   | 18.79   | 1.45   |
> |                     | PromptCoT-QwQ      | 1024     | 19.95   | 2.37   | 25.73   | 4.57   |
> |                     | ScaleQuest-Math    | 2035     | 14.66   | 2.24   | 23.34   | 5.39   |
> |                     | Conjecture-Prover  | 1164     | **20.67** | **3.76** | 27.11   | 7.08   |
> |                     | DExplorer (Staged) | 2340     | 19.66   | 3.31   | 29.21   | 8.02   |
> |                     | **DExplorer**      | **2726** |  *20.40*  |  *3.32*  | **30.39** | **8.14** |
> | **Difficulty**      | MUSTARD            | 3791     | -0.80   | 0.35   | 8.86    | 1.28   |
> |                     | PromptCoT-DS       | 457      | 11.12   | 2.38   | 11.12   | 2.38   |
> |                     | PromptCoT-QwQ      | 1024     | 12.35   | 2.37   | 19.16   | 3.90   |
> |                     | ScaleQuest-Math    | 2035     | 1.01    | 0.46   | 11.30   | 1.50   |
> |                     | Conjecture-Prover  | 1164     | **13.39** | 2.62   | 21.77   | 4.90   |
> |                     | DExplorer (Staged) | 2340     | 12.25   | 2.43   | 23.50   | 6.09   |
> |                     | **DExplorer**      | **2726** |  *13.12*  | **2.63** | **24.74** | **6.74** |
> | **Elegance**        | MUSTARD            | 3791     | 7.30    | 0.52   | 18.85   | 1.68   |
> |                     | PromptCoT-DS       | 457      | 15.71   | 1.90   | 15.71   | 1.90   |
> |                     | PromptCoT-QwQ      | 1024     | 17.07   | 2.16   | 24.00   | 3.38   |
> |                     | ScaleQuest-Math    | 2035     | 7.18    | 0.59   | 18.59   | 1.68   |
> |                     | Conjecture-Prover  | 1164     | **18.68** | **2.88** | 27.18   | 5.32   |
> |                     | DExplorer (Staged) | 2340     | 17.46   | 2.65   | 29.07   | 6.58   |
> |                     | **DExplorer**      | **2726** |  *18.15*  |  *2.79*  | **30.16** | **7.09** |
> | **Interestingness** | MUSTARD            | 3791     | -4.87   | 0.13   | 2.33    | 1.01   |
> |                     | PromptCoT-DS       | 457      | 8.40    | **2.23** | 8.40    | 2.23   |
> |                     | PromptCoT-QwQ      | 1024     | **9.01**  |  *2.07*  | 15.73   | 3.23   |
> |                     | ScaleQuest-Math    | 2035     | -2.31   | 0.27   | 6.20    | 1.09   |
> |                     | Conjecture-Prover  | 1164     | 8.50    | 1.83   | 17.00   | 3.34   |
> |                     | DExplorer (Staged) | 2340     | 7.71    | 1.79   | 19.15   | 4.25   |
> |                     | **DExplorer**      | **2726** |  *8.76*   | 1.97   | **20.53** | **4.79** |
>
> DExplorer significantly outperforms all baselines in the top-500 regime across most metrics, particularly in *Elegance* and *Interestingness*, demonstrating its capability to discover mathematically meaningful problems.
> While our overall scores are occasionally lower than Conjecture-Prover, this is expected as DExplorer generates a vastly larger volume of valid statements ($2726$ vs. $1164$), including simpler introductory lemmas that naturally dilute the average. However, the high scores in the Top-500 set confirm that DExplorer successfully pushes the frontier of mathematical discovery.

---

> ### Author Response · Authors · 2025-11-30
>
> > Weakness 4.1. The system still depends on Lean, Aesop, and LLM-based provers for verification, and about 3% of outputs fail the final check.
>
> > Question 4.1. Since the system still depends on **Lean, Aesop, and LLM-based provers** for validation, could the authors clarify what they mean by “eliminates reliance”?
>
> > Question 4.2. Would it be more accurate to describe the approach as “**reducing**” rather than “eliminating” dependency on external provers?
>
> R8. Thank you for this suggestion. We agree that "eliminating reliance" requires nuance. Our claim refers to the problem generation process. Unlike baselines like `Conjecture-Prover` that require a separate, often expensive, and fallible LLM prover to verify the provability of generated statements, DExploration can produce a proof alongside the statement. While DExploration relies on the Lean kernel to check proofs and execute tactics, it does not rely on external LLM provers to find the proof.
>
> However, we acknowledge that the final evaluation requires SOTA LLM-based provers to check for contradiction (*Contra.*) and measure the proof complexity (*Cplx.*). We have updated the term "eliminating" to "reducing" in the revision.
>
> > Weakness 5. the paper doesn’t include a **limitations section**.
> > Question 5.1. Could the authors discuss current **limitations**, such as tactic reliability, scalability, or remaining failure modes?
>
> R9. Certainly! We have now incorporated a "Limitations and Future Works" section in Appendix I, which covers:
> 1. **Controllable and Meaningful Generation.** Current DExploration is bootstrapped by supervised fine-tuning (SFT) on data distilled from existing theorem-proof pairs. While our experiments show that it generates problems with superior elegance and difficulty compared to baselines, it relies on implicit imitation rather than explicit optimization. Consequently, it underperforms at controlling specific domains, knowledge points, or difficulty levels on demand. Future works may incorporate conditional training variables or reinforcement learning (RL) with rubric-based reward functions to achieve precise control and further maximize the meaningfulness of the generation.
> 2. **Generalization to Out-of-distribution Domains.** Generalist LLMs are pretrained on web-scale knowledge, whereas the DExplorer is fine-tuned to explore mathematics primarily within the distribution of the training set. Future work may investigate techniques to maintain the broad knowledge coverage of pretrained LLMs while adapting them to formally verified deductive exploration.
> 3. **Co-evolution of Formal and Informal Reasoning.** DExploration effectively generates complex and difficult problems. This creates an opportunity for a virtuous cycle: using synthesized problems to adversarially train and improve formal provers or informal reasoners. Future work may explore this co-evolutionary pipeline to simultaneously push the boundaries of problem generation and automated reasoning.
>
> > Question 5.2. What are the **main barriers** to achieving a fully formalised notion of “deductive tactics”?
>
> R10. Thank you for this detailed comment. As revision in R1, Line 1007 in Appendix A was a finished TODO and was accidentally uncommented before our last LaTeX compilation. We sincerely apologize for any misunderstanding caused. Actually, the definition of deductive tactics is in lines 228-230. We have polished Appendix A in the revised version.
>
>
> [1] Autoformalization with Large Language Models
>
> [2] Introduction to classical and modern test theory

---

### Official Review · Reviewer_8iAd · 2025-10-31

**Soundness:** 2
**Presentation:** 3
**Contribution:** 3
**Rating:** 4
**Confidence:** 4

**Summary:**

This paper introduces DExploration, a Lean 4–based framework that reformulates mathematical problem synthesis as a step-by-step deductive exploration process.
Rather than directly generating conjectures or proofs, DExploration enables an agent to verifiably explore the mathematical environment through three atomic actions—Introduce, Deduce, and Submit—thereby generating both a formal statement and its proof in one unified loop.

**Strengths:**

1. The paper makes good contribution by reframing theorem synthesis as verifiable exploration. The Exploratory Transformation pipeline that reuses existing theorem–proof data to construct fine-grained exploration trajectories is a natural and scalable way to create augmented theorem proving dataset.

2. The authors perform a extensive ablation study that clearly supports their design choices, and the observed token efficiency improvement is both large and intuitively meaningful.

**Weaknesses:**

My main concern is the lack of external evaluation on public benchmarks. The experiments primarily analyze synthetic metrics (success rate, token cost, validity) on self-generated data. It remains unclear whether training on Exploratory Transformation data improves downstream theorem proving performance on public benchmarks such as MiniF2F, ProofNet or PutnamBench. Metrics such as “complexity” and “difficulty” are proxy measures based on proof length and model accuracy; these are helpful but insufficient for assessing real mathematical value.

**Questions:**

1. Have the authors tried fine-tuning a baseline prover (e.g., Goedel-Prover or DeepSeek-Prover) on the generated trajectories and evaluating it on public Lean benchmarks?
2. Could the authors clarify which sources (Mathlib, NuminaMath, etc.) were used to generate the training data, and how they ensured no overlap with test theorems?

---

> ### Author Response · Authors · 2025-11-30
>
> We greatly appreciate your constructive comments. We sincerely hope the following paper updates (U) and responses (R) align with your suggestions.
>
> - U1. Added "Improving theorem proving performance" (R1) to Appendix I (Limitations and Future Work).
> - U2. Added rubrics-based reward-model and LLM-as-a-judge evaluation to Appendix D.
>
> > Weakness 1 & Question 1. Effectiveness of theorem proving performance improvement.
>
> R1. Thank you for this crucial question. TLDR: Our method significantly improves theorem proving performance via Reinforcement Learning (RL).
>
> Given the relatively small size (2726 generated statements), we use verl[1] to train `DSP-V1.5` (DeepSeek-Prover-V1.5-RL, non-CoT) and `DSP-V2` (DeepSeek-Prover-V2-7B, non-CoT) using GRPO for 3 epochs. The hyperparameters follow Kimina Prover[2]. The results (`+RL` denotes RL applied to base models) show consistent improvements across benchmarks:
>
>
> | Bench        | Exp                         | Pass@1     | Pass@4     | Pass@8     | Pass@32    |
> | ------------ | --------------------------- | ---------- | ---------- | ---------- | ---------- |
> | MiniF2F-test | DSP-V1.5-7B                 | 4.92%      | 10.25%     | 15.98%     | 31.97%     |
> |              | **DSP-V1.5-7B + RL**        | **29.92%** | **39.34%** | **42.62%** | **46.31%** |
> |              | DSP-V2-7B (nonCoT)          | 50.82%     | 61.89%     | 62.30%     | 64.34%     |
> |              | **DSP-V2-7B (nonCoT) + RL** | **58.61%** | **63.52%** | **64.75%** | **66.39%** |
> | ProofNet     | DSP-V1.5-7B                 | 3.50%      | 7.55%      | 9.43%      | 12.94%     |
> |              | **DSP-V1.5-7B + RL**        | **8.36%**  | **10.24%** | **11.59%** | **14.29%** |
> |              | DSP-V2-7B (nonCoT)          | 15.36%     | 20.49%     | 21.29%     | 22.37%     |
> |              | **DSP-V2-7B (nonCoT) + RL** | **19.95%** | **21.56%** | **22.64%** | **22.91%** |
> | PutnamBench  | DSP-V1.5-7B                 | 0          | 1          | 1          | 1          |
> |              | DSP-V1.5-7B + RL            | **2**      | **4**      | **4**      | **6**      |
> |              | DSP-V2-7B (nonCoT)          | **6**      | **6**      | 6          | **9**      |
> |              | DSP-V2-7B (nonCoT) + RL     | 5          | **6**      | **7**      | 8          |
>
>
> We noted that the reproduction of `DSP-V1.5` underperformed compared to reported in the original paper. Error analysis reveals that the base model frequently outputs natural language (`NL Output`) instead of formal proof, likely due to chat template mismatches in verl. RL training significantly corrects this behavior.
>
> | Error Type | DSP-V1.5 | DSP-V1.5 + RL |
> | ---------- | -------- | ------------- |
> | NL Output  | 11294    | 715           |
> | Others     | 5378     | 14051         |
>
> Moreover, RL training significantly lowers the error of `Banned Tactic` (i.e., tactics including `sorry`, `apply?`, `hint` will directly close the proof without actually generating a proof). It also improves models' coherence with Mathlib and the Lean typing system by reducing `Hallucination of Objects` and `Type Error`.
>
> | Experiment               | DSP-V1.5 | DSP-V1.5 + RL |
> | ------------------------ | -------- | ------------- |
> | Banned Tactic            | 20.29%   | 12.49%        |
> | Hallucination of Objects | 26.53%   | 25.50%        |
> | Tactic Failure           | 16.59%   | 25.42%        |
> | Type Error               | 22.05%   | 20.08%        |
> | Failed to Close Goal     | 4.59%    | 6.71%         |
> | Syntax Error             | 8.39%    | 8.36%         |
> | Ambiguity Error          | 1.26%    | 1.23%         |
> | Timeout                  | 0.30%    | 0.21%         |

---

> ### Author Response · Authors · 2025-11-30
>
> > ... training on Exploratory Transformation data ...
>
> > ... fine-tuning a baseline prover ... on the generated trajectories ...
>
> To clarify, trajectories generated by exploratory transformation and DExploration are problem-generation steps that consist of *introducing* new variables/hypotheses, *deducing* new facts, and *submitting* one derived fact as a conclusion.
>
>
> We transformed the trajectories of all 2726 valid samples into SFT data by by removing *introducing*-steps and replacing *submitting*- steps with `exact`. Then, we fine-tuned DeepSeek-Prover-V2 into 1) a whole-proof generation model (`WG`) and 2) a proof search model (`PS`, following BFS-Prover[3]). Fine-tuning hyperparameters are identical to those in Appendix D.1.
>
> | Exp | MiniF2F-test | ProofNet |
> | --- | ------------ | -------- |
> | WG  | 62.30%       | 15.90%   |
> | PS  | 53.28%       | 15.90%   |
>
> Both fine-tuned models underperform the original `DeepSeek-Prover-V2`. This is expected since:
>
> 1. DExploration trajectories are designed for problem generation instead of theorem proving. *Introducing* steps uses `sorry`-tactics; and the purely deductive steps do not modify the proving target, preventing the model from learning powerful automations like `linarith`.
>
> 2. The dataset size (2726) is relatively small for SFT, leading to pontential overfitting.
>
> To sum up, RL on the generated problems improves the theorem proving performance. SFT on the problem generation trajectories results in inferior performance. We will add this aspect into Appendix I (Limitations and Future Work) and continue to explore it.
>
> Thank you again for this insightful comment :)

---

> > ### Author Response · Authors · 2025-11-30
> >
> > > Weakness 2. Proxy measures such as “complexity” and “difficulty” are helpful but insufficient for assessing real mathematical value.
> >
> > R2. This constructive comment aligns with Reviewers kBGE and ZhGX. In short, across evaluation settings, our method shows supeiror or on-par performance to all baselines and ablations on the mathematical value of the generated problems. This evaluation is added to Appendix D,
> >
> > We conducted a rubrics-based evaluation on *clarity*, *difficulty*, *elegance*, and *interestingness*.
> > We employ a state-of-the-art reward model (RM), `Skywork-Reward-V2-Llama-3.1-8B-40M`, and a generalist LLM, `DeepSeek-V3.2-Exp`, as judges to reduce bias. The LLM judges score the statements on a scale of 0-10 based on detailed rubrics provided in Appendix G.6. **Bold** highlights the best values, and *italic* indicates the second best values.
> >
> > | Metric              | Method             |          |  Overall  |          |  Top-500  |          |
> > | ------------------- | ------------------ | -------- | :-------: | :------: | :-------: | :------: |
> > |                     |                    | Valid    |    RM     |   LLM    |    RM     |   LLM    |
> > | **Clarity**         | MUSTARD            | 3791     |   13.33   |   2.95   |   21.97   |   7.47   |
> > |                     | PromptCoT-DS       | 457      |   18.79   |   1.45   |   18.79   |   1.45   |
> > |                     | PromptCoT-QwQ      | 1024     |   19.95   |   2.37   |   25.73   |   4.57   |
> > |                     | ScaleQuest-Math    | 2035     |   14.66   |   2.24   |   23.34   |   5.39   |
> > |                     | Conjecture-Prover  | 1164     | **20.67** | **3.76** |   27.11   |   7.08   |
> > |                     | DExplorer (Staged) | 2340     |   19.66   |   3.31   |   29.21   |   8.02   |
> > |                     | **DExplorer**      | **2726** |  *20.40*  |  *3.32*  | **30.39** | **8.14** |
> > | **Difficulty**      | MUSTARD            | 3791     |   -0.80   |   0.35   |   8.86    |   1.28   |
> > |                     | PromptCoT-DS       | 457      |   11.12   |   2.38   |   11.12   |   2.38   |
> > |                     | PromptCoT-QwQ      | 1024     |   12.35   |   2.37   |   19.16   |   3.90   |
> > |                     | ScaleQuest-Math    | 2035     |   1.01    |   0.46   |   11.30   |   1.50   |
> > |                     | Conjecture-Prover  | 1164     | **13.39** |   2.62   |   21.77   |   4.90   |
> > |                     | DExplorer (Staged) | 2340     |   12.25   |   2.43   |   23.50   |   6.09   |
> > |                     | **DExplorer**      | **2726** |  *13.12*  | **2.63** | **24.74** | **6.74** |
> > | **Elegance**        | MUSTARD            | 3791     |   7.30    |   0.52   |   18.85   |   1.68   |
> > |                     | PromptCoT-DS       | 457      |   15.71   |   1.90   |   15.71   |   1.90   |
> > |                     | PromptCoT-QwQ      | 1024     |   17.07   |   2.16   |   24.00   |   3.38   |
> > |                     | ScaleQuest-Math    | 2035     |   7.18    |   0.59   |   18.59   |   1.68   |
> > |                     | Conjecture-Prover  | 1164     | **18.68** | **2.88** |   27.18   |   5.32   |
> > |                     | DExplorer (Staged) | 2340     |   17.46   |   2.65   |   29.07   |   6.58   |
> > |                     | **DExplorer**      | **2726** |  *18.15*  |  *2.79*  | **30.16** | **7.09** |
> > | **Interestingness** | MUSTARD            | 3791     |   -4.87   |   0.13   |   2.33    |   1.01   |
> > |                     | PromptCoT-DS       | 457      |   8.40    | **2.23** |   8.40    |   2.23   |
> > |                     | PromptCoT-QwQ      | 1024     | **9.01**  |  *2.07*  |   15.73   |   3.23   |
> > |                     | ScaleQuest-Math    | 2035     |   -2.31   |   0.27   |   6.20    |   1.09   |
> > |                     | Conjecture-Prover  | 1164     |   8.50    |   1.83   |   17.00   |   3.34   |
> > |                     | DExplorer (Staged) | 2340     |   7.71    |   1.79   |   19.15   |   4.25   |
> > |                     | **DExplorer**      | **2726** |  *8.76*   |   1.97   | **20.53** | **4.79** |
> >
> > DExplorer significantly outperforms all baselines in the top-500 regime across most metrics, particularly in *Elegance* and *Interestingness*, demonstrating its capability to discover mathematically meaningful problems.
> > While our overall scores are occasionally lower than Conjecture-Prover, this is expected as DExplorer generates a vastly larger volume of valid statements ($2726$ vs. $1164$), including simpler introductory lemmas that naturally dilute the average. However, the high scores in the Top-500 set confirm that DExplorer successfully pushes the frontier of mathematical discovery.

---

> ### Author Response · Authors · 2025-11-30
>
> > Question 2. which sources (Mathlib, NuminaMath, etc.) were used to generate the training data, and how they ensured no overlap with test theorems
>
> R3. As described in line 306, we transform 39,509 formal statement-proof pairs in NuminaMath-Lean[4] into problem generation trajectories for fine-tuning DExplorer. It is a subset of the NuminaMath 1.5 dataset, which spans from Chinese high school math exercises to US and IMO competition problems.
>
> We use `Qwen/Qwen3-Embedding-8B` to embed all theorems in the benchmarks and the statements in NuminaMath-Lean, and have examined the semantic equivalence between each test theorem and the most similar statement in NuminaMath-Lean. Results show that 60/244 of MiniF2F-test, 1/371 of ProofNet, and 12/642 of PutnamBench are contaminated.
>
> We then evaluated the semantic similarity between the 2726 training set and all benchmarks. All similarities are $<0.9$. Manual examination found 4/244 of MiniF2F-test, 1/371 of ProofNet, and 0/642 of PutnamBench are contaminated. All overlaps are already proven by `DSP-V2`. Therefore, the results of `DSP-V2` in [R1] (improving theorem proving performance) are relatively fair.
>
> Thank you again for this important and insightful suggestion! We will emphasize dataset decontamination in future works on using DExplorer to improve LLM provers.
>
> [1] verl: Volcano Engine Reinforcement Learning for LLMs
>
> [2] Kimina-Prover Preview: Towards Large Formal Reasoning Models with Reinforcement Learning
>
> [3] BFS-Prover: Scalable Best-First Tree Search for  LLM-based Automatic Theorem Proving
>
> [4] https://huggingface.co/datasets/AI-MO/NuminaMath-LEAN
>
> [5] NuminaMath: The largest public dataset in AI4Maths with 860k pairs of competition math problems and solutions

---

### Author Response · Authors · 2025-12-03

Dear Area Chairs,

In light of the recent unforeseen incident, we fully acknowledge the challenges facing the ICLR community and the additional responsibilities you now bear.

We sincerely thank you and the reviewers for your time, feedback, and constructive suggestions. To facilitate your workflow, a brief summarization of the discussion phase is provided below:

In this work, we propose 1) **DExploration** _(**D**eductive **Exploration**)_, formulating problem synthesis as a whole-process verifiable exploration process rather than one-shot generation; and 2) **Exploratory Transformation**, which transforms existing large-scale theorem-proving corpora into exploration trajectories to train DExplorer. Experiments validate DExplorer's effectiveness across multiple aspects.

Reviewers characterized the paper as tackling one of the hardest open challenges in AI4Math (`ZhGX`); DExploration as genuinely impressive (`ZhGX`) and closely mirroring the workflow of mathematicians (`kBGE,ZhGX`); Exploratory Transformation as natural and scalable (`8iAd`), well-founded (`kBGE`), clever and principled (`ZhGX`); and the experiments as extensive (`8iAd`), thorough and convincing (`ZhGX`) with notable improvements (`kBGE`).

Their main comments/questions and our responses are as follows:
- **More metrics for assessing mathematical value** (`8iAd,kBGE,ZhGX`): We added a rubric-based evaluation on *clarity*, *difficulty*, *elegance*, and *interestingness* using a SOTA reward model and a SOTA LLM to Appendix D. DExplorer shows superior or on-par performance compared to all baselines and ablations.
- **Failure analysis of samples that failed to reconstruct** (`kBGE,ZhGX`): All issues were resolved after careful debugging. Error modes include 1) failure to transform pretty-printed `Expr` back to `Syntax`; 2) deducing steps introducing new variables; 3) introducing steps anonymizing existing variables; and 4) tactics behaving differently between problem generation and proof reconstruction.
- **Improving theorem proving (TP) performance** (`8iAd`): We demonstrated that TP performance is significantly improved via reinforcement learning on the 2726 generated data.
- **Data source for DExplorer training** (`8iAd`): Provided in Line 306 (NuminaMath-Lean).
- **Definition of "deductive tactics"** (`kBGE`): Provided in Lines 228-230.
- **Proof of the provability of statements generated by DExploration** (`kBGE`): Added to Appendix A. TL;DR: Encapsulation ensures tactic execution contexts are identical between proof reconstruction and problem generation.
- **Sensitivity to the choice of prover or verification strategy** (`kBGE`): DExplorer without the lightweight explosion check still significantly outperforms baselines. Regarding evaluation, $K=3\times 4$ already saturates the contradiction check.
- **Why only 100 samples for manual evaluation** (`kBGE`): All generated problems are guaranteed to be provable. Manual evaluation targets informalization quality, which is direct. We checked 400 additional samples, resulting in even higher accuracy.
- **System still depends on Lean, Aesop, and LLM-based provers for verification** (`kBGE`): Lean functions only as a verifier; DExplorer does not rely on Aesop or LLM-based provers to prove generated statements, as it produces a proof alongside exploration.
- **Absence of a limitations section** (`kBGE`): We added a "Limitations and Future Works" section (Appendix I), covering: 1) Controllable and Meaningful Generation; 2) Generalization to Out-of-distribution (OOD) Domains; and 3) Co-evolution of Formal and Informal Reasoning.
- **Will exploration quality degrade if generalization to OOD domains** (`ZhGX`): Yes, it degrades noticeably. This is now discussed in the Limitations section.
- **Guiding exploration toward specific difficulty levels** (`ZhGX`): We found that limiting the minimum steps before submitting improves complexity and rubric-based metrics. Finer-grained difficulty controlling is discussed in Future Work.
- **Sweet spot in quality-to-time tradeoff** (`ZhGX`): Discussed in Sec. 5.3.1 & Appendix C. $N_s=8$ is optimal if focusing on token cost; $N_s=16, 32$ are better when considering complexity.
- **Can DExplorer result in a self-improving cycle** (`ZhGX`): Yes. We showed that RL on generated statements improves TP performance. Further validation is left to future work due to time and resource constraints.

Sincere thanks again to the reviewers. Their invaluable feedback has significantly improved this paper. We hope this "both theoretically elegant and practically useful" work, which "closely mirrors the workflow of mathematicians," can help "tackle one of the hardest open challenges in AI4Math."

Moreover, we fully recognize the challenges faced by the ICLR community and the additional workload imposed on the ACs. We affirm that we have strictly adhered to the ICLR Code of Ethics.

---

### Meta-Review · Area_Chair_cd49 · 2026-01-10

**Summary:**

-	Summary:

This paper reformulates mathematical problem synthesis as a step-by-step deductive exploration process rather than one-shot generation, and experiments show its effectiveness on reasoning and other tasks.

-	Reviewer Concerns:

The main concerns raised by reviewers included: (1) potentially simplistic Lean tactics, (2) limited sample size, (3) insufficient validation metrics, and (4) uncertain generalizability to other open datasets.

-	Author Response:

During the discussion period, the authors adequately addressed these concerns through additional experimental evidence and explicit limitation statements.

-	Assessment:

This paper presents a novel incremental approach to mathematical problem synthesis, contrasting with traditional single-round generation methods. The Lean 4-based framework ensures provability of synthesized problems, addressing critical issues of data exhaustion, contamination, and leakage in mathematical reasoning datasets.

-	Recommendation:

After careful consideration, I recommend accepting this paper as it makes meaningful contributions to mathematical problem-solving in AI reasoning.

**Reviewer Concerns:**

I think the Reviewer concerns of 8iAd, kBGE, ZhGX are addressed by the rebuttal.

**Reviewer Scores:**

•  Reviewer 8iAd: Increased rating to 6 (main concerns resolved with experimental support)

•  Reviewer kBGE: Increased rating to 6 (main concerns resolved with experimental support)

•  Reviewer ZhGX: Maintained all original scores

---

### Decision · Program_Chairs · 2026-01-26

Accept (Poster)